# LEARNING MODEL UNCERTAINTY AS VARIANCE-MINIMIZING INSTANCE WEIGHTS

**Nishant Jain, Karthikeyan Shanmugam & Pradeep Shenoy**
Google Research India
`{nishantjn, karthikeyanvs, shenoypradeep}@google.com`

## ABSTRACT

Predictive uncertainty–a model's self-awareness regarding its accuracy on an input–is key for both building robust models via training interventions and for test-time applications such as selective classification. We propose a novel instance-conditional reweighting approach that captures predictive uncertainty using an auxiliary network, and unifies these train- and test-time applications. The auxiliary network is trained using a meta-objective in a bilevel optimization framework. A key contribution of our proposal is the meta-objective of minimizing dropout variance, an approximation of Bayesian predictive uncertainty, We show in controlled experiments that we effectively capture diverse specific notions of uncertainty through this meta-objective, while previous approaches only capture certain aspects. These results translate to significant gains in real-world settings–selective classification, label noise, domain adaptation, calibration–and across datasets–Imagenet, Cifar100, diabetic retinopathy, Camelyon, WILDs, Imagenet-C,-A,-R, Clothing1M, etc. For Diabetic Retinopathy, we see upto 3.4%/3.3% accuracy  AUC gains over SOTA in selective classification. We also improve upon large-scale pretrained models such as PLEX (Tran et al., 2022).

## 1 INTRODUCTION

In applications with significant cost of error, classifiers need to accurately quantify and communicate uncertainty about their predictions. Test-time applications of uncertainty include OOD detection (Hendrycks & Gimpel, 2017), and selective classification (rejecting test instances where the model is not confident (El-Yaniv et al., 2010)). Intrinsic classifier measures of uncertainty (softmax probabilities, or derived metrics) perform poorly compared to more sophisticated approaches–e.g., methods that learn an extra "reject" category at train time (Liu et al., 2019), or estimate the Bayesian posterior predictive uncertainty given a distribution over model weights (Gal & Ghahramani, 2016). Other work show benefits of reweighting training instances or reweighting subpopulations within the data improving generalization (Zhou et al., 2022a; Faw et al., 2020; Ghosh et al., 2018). The intuition is that the weights realign the training distribution to simulate a possibly different, or worst-case target distribution (see e.g., Kumar et al. (2023).

In this work, we are interested in learning an instance dependent weight function (on the training set) that captures predictive uncertainty. A significant challenge is that a wide range of underlying causal factors can contribute to uncertainty such as input-dependent label noise, missing features, and distribution shift between training and test data (refer Sec. 4 for more details and analysis). For covariate shift between train and test, weights on training set related to importance sampling or kernel mean matching (Sugiyama et al., 2007; Yu & Szepesvári, 2012) are appropriate. Other neural net based solutions transform train to match test in distribution (Ganin et al., 2016). In the case of label noise that varies across different regions in a domain, it is known that downweighing samples with larger uncertainty (Das et al., 2023) is the best solution. In robust optimization, one tries to weigh samples that yields the worst loss (Levy et al., 2020) in an uncertainty ball around the current distribution. Of the related literature, no work, to the best of our knowledge, addresses all these diverse sources of uncertainty.

The question that motivates our work is: *Given training and validation set, what is the best reweighing function of training instances that yields a good uncertainty measure at test time? Further, what robustness properties are achieved by the reweighted classifier?*

We propose a novel instance dependent weight learning algorithm for learning predictive uncertainty – REVAR(**Re**weighting for Dropout **Va**riance **R**eduction). We propose to learn an auxiliary uncertainty model $p = g(x)$ (which we call U-SCORE) alongside training of the primary model $y = f(x)$. U-SCORE unifies train and test-time applications of uncertainty. Our primary algorithmic insight is the use of a novel dropout based variance regularization term in the U-SCORE objective. Below we summarize our approach and key contributions.

**1.** U-SCORE **as an uncertainty measure:** We learn an instance-conditional function $p = g(x)$, allowing us to capture a rich notion of model uncertainty. This function is learned in a nested or bi-level optimization framework where the classifier minimizes a weighted training loss, and the U-SCORE minimizes a *meta-loss* from the resulting classifier on a separate meta-training dataset. Our approach strictly generalizes previous reweighting approaches based on bi-level optimization (Shu et al., 2019; Zhang & Pfister, 2021), as they cannot be used at test-time. We propose a variance reduction meta-regularization loss that forms a key part of our approach, and incentivizes the learned function $w = g(x)$ to faithfully capture uncertainty as a function of the input instance.

**2.** U-SCORE **scaling with different uncertainty sources:** For robust test-time prediction, a reweighting approach should ideally downweight training samples with high label noise (since this would be independent uncertainty) but emphasize (upweight) hard examples in terms of overlap with respect to validation data. We demonstrate through controlled synthetic linear regression experiments, that U-SCORE scores achieve both these desirable behaviors and even smoothly interpolates between two different scaling behaviors when different sources of uncertainty are present. Ablations show that changes to our meta-loss does not yield the same performance.

**3. Real-world applications:** U-SCORE outperforms conventional and state-of-the-art measures of predictive uncertainty by significant margins in a wide range of datasets (Diabetic Retinopathy, CIFAR-100, ImageNet, Clothing1M, etc) and domain shift conditions (Camelyon, WILDS, Imagenet-A,-C,-R, etc). These results mirror and strengthen the findings of the controlled study in real-life applications. As an example, in Diabetic Retinopathy, a well-studied benchmark dataset for selective classification, we show upto 3.4%/3.3% accuracy & AUC gains over state-of-the-art methods in the domain shift setting. We also improve upon large-scale pretrained models such as PLEX (Tran et al., 2022) by $\sim$4% relative on label uncertainty and from 7.5 to 6.2 ECE in calibration.

## 2 RELATED WORK

**Uncertainty Estimation.** Uncertainty in deterministic neural networks is extensively studied, e.g., via ensemble modelling (Wen et al., 2020; Valdenegro-Toro, 2019; Lakshminarayanan et al., 2017) or using prior assumptions (Oala et al., 2020; Możejko et al., 2018; Malinin & Gales, 2018; Sensoy et al., 2018) for calculating uncertainty in a single forward pass. Bayesian NNs offer a principled approach(Buntine, 1991; Tishby et al., 1989; Denker et al., 1987; Blundell et al., 2015; Kwon et al., 2020) by modeling a distribution over model weights. They output posterior distributions over predictions after marginalizing weight uncertainty, where the distribution spread encodes uncertainty. Gal & Ghahramani (2016) connects deterministic and Bayesian NNs by using multiple forward passes with weight dropout to approximate the posterior; several works (Kwon et al., 2018; Kendall & Gal, 2017; Kwon et al., 2020; Pearce et al., 2020) use this to quantify predictive uncertainty.

**Instance weighting and bilevel optimization.** Importance weighting of training instances is popular in robust learning, e.g., OOD generalization (Zhou et al., 2022a), handling concept drift (Jain & Shenoy, 2024), robustness to label noise (Shu et al., 2019; Ren et al., 2018; Zhang & Pfister, 2021; Sivasubramanian et al., 2023), Group Distributionally Robust Optimization (Faw et al., 2020; Mohri et al., 2019; Ghosh et al., 2018), covariate shift (Sugiyama et al., 2008). Weights could be a predefined function of margin or loss (Liu et al., 2021; Kumar et al., 2023; Sugiyama et al., 2008) or learned via bi-level optimization (Ren et al., 2018; Zhang & Pfister, 2021; Zhou et al., 2022a). In these latter cases, instance weights are free parameters (Ren et al., 2018), or a learned function of loss/instance (Shu et al., 2019; Holtz et al., 2021; Jain & Shenoy, 2024). Bilevel optimization is widely used in many settings, and can be solved efficiently (Bertrand et al., 2020; Blondel et al., 2022). Finally,

recent work has explored how to meta-optimize bilevel formulations by choosing the validation set used for the nested optimization (Jain et al., 2024).

**Formal models of uncertainty.** Uncertainty in neural network predictions can be decomposed into: (a) Uncertainty in input (*aleatoric*) and (b) uncertainty in model parameters (*epistemic*). Recent works (Kendall & Gal, 2017; Kwon et al., 2018; Smith & Gal, 2018; Zhou et al., 2022b; Depeweg et al., 2018; 2017; Hüllermeier & Waegeman, 2021) propose explicit models for these uncertainties in both classification and regression. (Kendall & Gal, 2017) model epistemic uncertainty for regression as the variance of the predicted means per sample, and for classification using the pre-softmax layer as a Gaussian Distribution with associated mean and variance. Kwon et al. (2018) directly calculate variance in the predictive probabilities. Depeweg et al. (2017) proposed an information-theoretic decomposition into epistemic and aleatoric uncertainties in reinforcement learning. Smith & Gal (2018) proposed Mutual Information between the expected softmax output and the estimated posterior as a measure of model uncertainty. Valdenegro-Toro & Mori (2022) aim to disentangle notions of uncertainty for various quantification methods and showed that aleatoric and epistemic uncertainty are related to each other, contradicting previous assumptions. Finally, recent work (Zhang et al., 2021) proposed a regularization scheme for robustness which also minimizes epistemic uncertainty.

**Selective Classification.** Selective classification offers a model the option of rejecting test instances, within a bounded rejection rate, to maximize accuracy on predicted instances. It is a benchmark for uncertainty measures in Bayesian NNs (Filos et al., 2019). Many approaches have been studied, including risk minimization with constraints (El-Yaniv et al., 2010); optimizing selective calibration (Fisch et al., 2022); entropy regularization (Feng et al., 2022) and training-dynamics-based ensembles (Rabanser et al., 2022); a separate "reject" output head (Deep Gamblers (Liu et al., 2019)); a secondary network for rejection (Selective Net (Geifman & El-Yaniv, 2019)); optimizing training dynamics (Self-adaptive training (SAT (Huang et al., 2020)); class-wise rejection models (OSP (Gangrade et al., 2021)); variance over monte-carlo dropout predictions (MCD (Gal & Ghahramani, 2016)). We compare against these and a range of other competitive baselines in our experiments.

# 3 ReVaR: DROPOUT VARIANCE REDUCTION

## 3.1 Preliminaries & objective

Given a dataset $D = (X, Y)$ with input-output pairs $X \in \mathcal{X}, Y \in \mathcal{Y}$, supervised learning aims to learn a model $f_\theta : \mathcal{X} \mapsto \mathcal{Y}$, where $\theta$ denote the model parameters. In this work, we wish to also learn a measure of *predictive uncertainty* $g_\Theta : \mathcal{X} \mapsto [0, 1]$ (with $\Theta$ denoting the parameters) for our classifier $f_\theta$, as a function of the input $x$. The function $g(x)$ should output high scores on inputs $x$ for which $f(\cdot)$ is *most uncertain*, or is more likely to be incorrect. A well-calibrated measure of model uncertainty $g(x)$ could be used for many applications in failure detection & avoidance – for instance, in selective classification (El-Yaniv et al., 2010), one can abstain from making a prediction by applying a suitable threshold $\lambda$ on $g(\cdot)$ in the following manner:

$$(f, g)(x) = \begin{cases} f(x) & \text{if } g(x) < \lambda; \\ \text{can't predict} & \text{if } g(x) \geq \lambda; \end{cases} \tag{1}$$

An effective measure $g(\cdot)$ should maximize accuracy of $f(\cdot)$ on unrejected test instances given a targeted rejection rate. Similarly, one could also evaluate $g(\cdot)$ by checking its calibration; i.e., whether the rank-ordering of instances based on $g$ correlates strongly with the accuracy of the classifier $f$.

In our work, we also use a *specialized set* $(X^s, Y^s)$ (also referred to as a validation set), separate from the training set $(X, Y)$ of the classifier, to obtain an unbiased measure of classifier performance. Where available, we use a validation set that is representative of the test data; as in previous work, this helps our learned classifier to better adapt to test data in cases of distribution shifts.

## 3.2 The learned reweighting framework

We start with a bilevel optimization problem for *learned reweighting* of a supervised objective:

$$\theta^* = \arg\min_\theta \frac{1}{N} \sum_{i=1}^{N} w_i \cdot l_{train}(y_i, f_\theta(x_i)) \quad s.t. \ \{w_i^*\} = \arg\min_{\{w_i\}} \sum_{j=1}^{M} l_{meta}(x_j^s, y_j^s, \theta^*) \tag{2}$$

where $x_i \in X$, $y_i \in Y$ and $x_i^s \in X^s$, $y_i^s \in Y^s$. Here, $(N, M)$ are the sizes of the train and validation sets, $\theta$ are model parameters for $f$, $w_i$ are free parameters reweighting instance losses, and $(l_{train}(), l_{meta}())$ are suitably chosen loss functions (e.g., cross-entropy). The meta-loss above is an *implicit function* of the weights $\{w_i^*\}$, through their influence on $\theta^*$. The formulation finds a (model, weights) pair such that the weighted train loss, and the unweighted validation loss for the learned $f(\cdot)$, are both minimized.

**Intuition**: Bilevel optimization has been used for overcoming noisy labels in training data (Ren et al., 2018), and in addressing covariate shift at instance level (Sugiyama et al., 2007) and group level (Mohri et al., 2019). In many of these applications, as well as in general for improved model generalization (Kumar et al., 2023), there is theoretical and empirical evidence that the training instance weights should be proportional to instance hardness, or uncertainty under covariate shift. Thus, the learned reweighting objective is a good starting point for capturing instance uncertainty, at least with respect to the training data.

### 3.3 INSTANCE-CONDITIONAL WEIGHTS IN REVAR

Our goal in REVAR is to learn an instance-conditional scorer U-SCORE that is used both for train-time reweighting, and at test time as a measure of predictive uncertainty. Previous work on bilevel optimization for reweighting (Shu et al., 2019; Ren et al., 2018; Zhang & Pfister, 2021) cannot be used at test time because the learned weights are free parameters (Ren et al., 2018) or a function of instance loss (Shu et al., 2019; Zhang & Pfister, 2021). We address this challenge by learning instance weights as a direct function of the instance itself, i.e., $w = g_\Theta(x)$, allowing us to capture a much richer and unconstrained measure of model uncertainty, which is also robustly estimated using the bilevel formulation. Our bilevel formulation now becomes:

$$\theta^* = \arg\min_\theta \frac{1}{N} \sum_{i=1}^N g_\Theta(x_i) \cdot l(y_i, f_\theta(x_i)) \quad s.t. \ \Theta^* = \arg\min_\Theta \mathcal{L}_{meta}(X^s, Y^s, \theta^*) \tag{3}$$

where $\theta^*$, $\Theta^*$ correspond to optimal parameters for classifier, U-SCORE respectively.

### 3.4 VARIANCE MINIMIZATION AS META-REGULARIZATION

We now define the meta-loss $\mathcal{L}_{meta}$ and in particular, a novel variance-minimizing regularizer that substantially improves the ability of $g(\cdot)$ to capture model uncertainty. This meta-regularizer $l_{eps}(\theta, x)$ is added to the cross-entropy classification loss $l(y, f_\theta(x))$ that is typically part of the meta-loss, leading to the following meta-objective on the specialized set $(\mathcal{X}^s, \mathcal{Y}^s)$:

$$\mathcal{L}_{meta} = \mathcal{L}_c(X_s, Y_s) + \mathcal{L}_{eps}(\theta, X_s) = \sum_{j=1}^M l(y_j^s, f_\theta(x_j^s)) + l_{eps}(\theta, x_j^s) \tag{4}$$

**Minimizing Bayesian Posterior uncertainty:** We take inspiration from the Bayesian NN literature which regularizes the posterior on weight distribution so as to avoid overfitting or to embed extra domain knowledge. Unlike standard neural networks which output a point estimate for a given input, Bayesian networks (Buntine, 1991; Tishby et al., 1989; Denker et al., 1987; Blundell et al., 2015; Kwon et al., 2020) learn a distribution $p(\omega|D)$ over the neural network weights $\omega$ given the dataset $D$ using maximum a posteriori probability (MAP) estimation. The predictive distribution for the output $y^*$, given the input $x$ and $D$, can be then calculated by marginalisation as follows: $p(y^*|x^*, D) = \int p(y^*|x^*, \omega)p(\omega|D)d\omega \approx \frac{1}{K}\sum_{k=1}^K p(y^*|x^*, \omega^k)$. Here we utilize a recent result (Gal & Ghahramani, 2016) that augmenting the training of a deterministic neural network with dropout regularization yields a variational approximation for a Bayesian Neural Network. At test time, taking multiple forward passes through the neural network for different dropout masks yields a *Monte-Carlo* approximation to Bayesian inference, and thereby a predictive distribution. The variance over these Monte Carlo samples is therefore a measure of predictive uncertainty:

$$l_{eps}(\theta, x) \approx \frac{1}{K}\left(\sum_{k=1}^K (f_{\mathcal{D}_k \odot \theta}(x) - E[f_{\mathcal{D}_k \odot \theta}(x)])^2\right) \tag{5}$$

where $\mathcal{D}_k$ denotes the dropout mask at $k^{th}$ sample and $\mathcal{D}_k \odot \theta$ denotes the application of this dropout mask to the neural network parameters. This MCD measure is popular as an estimate of instance

uncertainty (Gal & Ghahramani, 2016), and is competitive with state-of-the-art methods for selective classification (Filos et al., 2019).

We propose to use this variance-based estimate of posterior uncertainty as a *meta-regularization term*, i.e., instead of minimizing the posterior uncertainty on the training data w.r.t. primary model parameters $\theta$, we minimize it w.r.t. U-SCORE parameters $\Theta$ on the specialized set instead.

**Intuition for meta-regularization:** We make two significant, intertwined technical contributions: (1) using an instance conditioning network $g_\Theta(x)$ for calculating training instance weights, and (2) using variance minimization in the meta-objective for $g(x)$. The network $g(x)$ can be applied to unseen test instances without labels, and enables test-time applications of uncertainty not possible with previous methods. For the variance minimization loss, note that we minimize MCD variance of the primary model $f(x)$ on the *validation set*; this is data not available to the primary model during training. This loss has to be indirectly minimized through training data reweightings, which in turn are accomplished through $g(x)$. In other words, the function $g(x)$ is forced to focus on the most uncertain/ most informative instances $x$ in the training data. Thus, our goal with the meta-regularizer is not, primarily, to build a classifier $f(x)$ with low output variance; we instead use it indirectly, via the meta-objective, to improve the quality of the uncertainty measure $g(x)$.

## 3.5 META-LEARNING WITH BILEVEL LOSS

The modeling choices we have laid out above result in a bi-level optimization scheme involving the meta-network and classifier parameters. This is because the values of each parameter set $\theta$ and $\Theta$ influence the optimization objective of the other. Expressing $\mathcal{L}_{meta}$ as a function $\mathcal{L}$ of inputs $X^s$, $Y^s$, $\theta$, this bi-level optimization scheme can be formalized as:

**Calculating updates**. Instead of solving completely for the inner loop (optimizing $\Theta$) for every setting of the outer parameter $\theta$, we aim to solve this bilevel optimization using alternating stochastic gradient descent updates. At a high level, the updates are:

$$\Theta_{t+1} = \Theta_t - \alpha_1 \nabla_\Theta \mathcal{L}(X^s, Y^s, \theta_t) \quad ; \quad \theta_{t+1} = \theta_t - \alpha_2 \frac{1}{N} \nabla_\theta \left( \sum_{i=1}^{N} g_\Theta(x_i) \cdot l(y_i, f_\theta(x_i)) \right) \quad (6)$$

where $\alpha_1$ and $\alpha_2$ are the learning rates corresponding to these networks, $l$ is the classification loss on the training dataset $(X,Y)$ using the classifier network with parameters $\theta$. This style of stochastic optimization is commonly used for solving bilevel optimization problems in a variety of settings (Algan & Ulusoy, 2020; Shu et al., 2019). Further details, including all approximations used for deriving these equations, are provided in the appendix. Also, the U-SCORE is implemented as a standard neural network architecture like the classifier (please refer appendix).

## 4 U-SCORE CAPTURES DIFFERENT SOURCES OF UNCERTAINTY

We now create a set of synthetic generative models for linear regression and study the performance of our algorithm for conceptual insights. We investigate three kinds of uncertainty that depends on the input instance $x$: 1) Samples that are atypical with respect to train but typical with respect to validation 2) Samples where label noise is higher 3) Samples where uncertainty in the label is due to some unobserved latent features that affect the label.

Usually (1) and (3) are considered to be "epistemic" uncertainty while (2) is understood as "aleatoric" uncertainty. (1) is due to covariate shift and (3) is due to missing features relevant for the label. Surprisingly, we show in this section that *our algorithm's weights are proportional to uncertainty from (1) while being inversely proportional to uncertainty of type (2) and (3)*. This is also desirable from a theoretical perspective, as we explain below–for instance, when (1) and (3) are absent, the best solution is to downweight samples with larger label noise (Das et al., 2023). Similarly, when only (1) is present, one would desire examples that are typical with respect to validation and atypical with respect to train to be weighted higher. Our algorithm captures these notions; furthermore it smoothly interpolates between them depending on the mix of different sources of uncertainty.

**Generative Model:** For all the results in this section, for both training and validation data for all , $Y$ is sampled as follows.

$$Y = W_{\text{data}}^T X + (\mathcal{N}(0,1) \cdot [c + G^T X]) \quad (7)$$

$X \in \mathbb{R}^{72 \times 1}$. $X = [X_o X_l], X_o \in \mathbb{R}^{48 \times 1}, X_l \in \mathbb{R}^{24 \times 1}$. For training data, we sample $X^{\text{train}} \sim \mathcal{N}(\mu, \Sigma)$. For validation, $X^{\text{val}} \sim \mathcal{N}(\mu', \Sigma)$ where $\mu' = \mu + s\mathcal{N}(\mu_s, \Sigma_s)$; here $s > 0$ is a scalar that determines the amount of covariate shift between training and validation. $W_{\text{data}}^T = [W_o^T \; W_l^T]$ where $W_o \in \mathbb{R}^{48 \times 1}$, $W_l \in \mathbb{R}^{24 \times 1}$.

**Intuition:** The above generative model[1] is chosen to allow testing of the different scenarios described above, w.r.t. label noise and covariate shift. We control additive noise via the variables $c$ (zero-mean noise) and $G$ (scaled input-dependent noise). We can introduce covariate shift between training and validation data using $s$. Additionally, although the output $Y$ depends on all components of $X$, we can consider scenarios where only a portion of $X := [X_o X_l]$ is made available to the learner ($X_o$ is *observed* while $X_l$ are *latent* variables that influence $Y$ but are unobserved). Finally, we can also create combined scenarios where more than one of these factors are at play.

**Evaluation, Baselines & Metrics:** We train on paired train-validation datasets sampled according to each scenario, and inspect U-SCORE scores for the training set. For each scenario, we predict a theoretical ideal for the instance dependent weights, and calculate $R^2$ score of model fits for the U-SCORE outputs against the theoretical ideal. We compare against MWN (Shu et al., 2019), which calculates *loss-dependent* instance weights using bilevel optimization. We also evaluate a second baseline that is identical to REVAR except for the meta-regularization term–we term this Instance-Based Reweighting (IBR).

Table 1: $R^2$ metric. $\lambda_1, \lambda_2$ are fitting coefficients. $h$ (hardness) is Euclidean distance from training data mean $h = (x - \mu)^2$, and captures magnitude of covariate shift. Other terms quantify sample dependent label noise.

| S | Target | MWN | IBR | Ours |
|---|---|---|---|---|
| 1 | $\frac{\lambda_1}{|G^T X|^2}$ | 0.77 | 0.78 | 0.84 |
| 2 | $\frac{\lambda_1}{|G^T X|^2} + \lambda_2 \cdot h$ | 0.58 | 0.62 | 0.80 |
| 3 | $\frac{\lambda_1}{W_l^T \Sigma(X_l|X_o)W_l}$ | 0.46 | 0.52 | 0.81 |
| 4 | $\frac{\lambda_1}{W_l^T \Sigma(X_l|X_o)W_l} + \lambda_2 \cdot h$ | 0.51 | 0.57 | 0.82 |
| 5 | $\lambda_1 \cdot \mathcal{U}(0,1)$ | 0.44 | 0.58 | 0.84 |

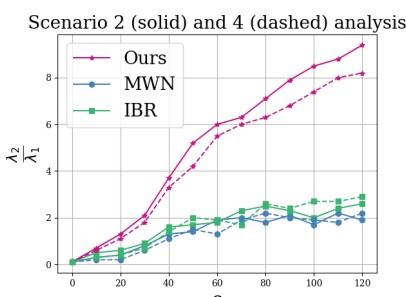

Figure 1: Scenario 2 and 4 analysis with increasing distribution shift

**Scenario 1 - Sample Dependent Label Noise and No Shift:**. $c = 0, s = 0, G \neq 0$. This scenario has no covariate shift, but label uncertainty in both train and validation depend on the sample. Label noise scales as $|G^T X|^2$, while weights of the meta-network are *inversely proportional* to this quantity; this weighting is shown to be optimal by recent theoretical work (Das et al., 2023)). (Tab. 1).

**Scenario 2 - Sample Dependent Label noise and Covariate Shift:** We set $c = 0, G \neq 0, s \neq 0$. We expect the weights to be inversely proportional to label noise (Scenario 1); we also expect weights to be directly proportional to the uncertainty due to covariate shift, i.e., $h := (x - \mu)^2$. This latter idea draws from classical work on importance sampling under covariate shift (Sugiyama et al., 2007) and many follow-on papers that theoretically motivate weighting training data proportional to hardness or uncertainty. When both factors are at play, we posit a simple linear combination $w(x) \sim \frac{\lambda_1}{|G^T x|^2} + \lambda_2 \cdot h$. as an *a priori* desirable weighting of training instances. Interestingly, REVAR weights in fact correlate very strongly with this linear model. Further, REVAR weights shift smoothly towards uncertainties from covariate shift as its magnitude increases Sec. 4.

**Scenario 3 - Hardness due to missing relevant features:** We set $c = 1, G = 0, s = 0$. However, only $X_o$ is available to the learner in both train and validation. Here, the missing or latent features $X_l$ influence the label in a way inaccessible to the learner. Interestingly this behaves much like sample-dependent label noise–given the features seen ($X_o$), the "label noise" (contribution of $X_l$ that cannot be modeled) is controlled by the conditional dependence of $X_l|X_o$, and is therefore proportional to $W_l^T \Sigma(X_l|X_o)W_l$. This is identical to scenario 1, and the optimal solution is again inverse weighting of instances. Indeed, the weights predicted by U-SCORE roughly scales as $\frac{1}{W_l^T \Sigma(X_l|X_o)W_l}$ (Tab. 1). Although conventionally treated as "epistemic uncertainty", our meta network's weights are inversely proportional to this, as desired.

---

[1]The exact dimensionalities are not materially relevant to our findings; we have also evaluated other settings with similar results.

Table 2: Selective classification: REVAR consistently scores highest on the metric Area under Accuracy-Rejection curve (Zhang et al., 2014), including on larger datasets such as ImageNet.

| | Selective Classification Baselines | | | | | New Baselines | | REVAR |
| | SR | MCD | DG | SN | SAT | VR | MBR | Ours |
|---|---|---|---|---|---|---|---|---|
| DR(In-Dist.) | $92.87 \pm 0.1$ | $93.44 \pm 0.0$ | $93.07 \pm 0.1$ | $93.13 \pm 0.1$ | $93.56 \pm 0.1$ | $92.55 \pm 0.1$ | $92.95 \pm 0.2$ | $\mathbf{94.12 \pm 0.1}$ |
| DR(OOD) | $87.67 \pm 0.1$ | $88.27 \pm 0.1$ | $88.07 \pm 0.2$ | $88.56 \pm 0.1$ | $88.97 \pm 0.2$ | $87.91 \pm 0.1$ | $88.06 \pm 0.3$ | $\mathbf{89.94 \pm 0.1}$ |
| CIFAR-100 | $92.30 \pm 0.1$ | $92.71 \pm 0.1$ | $92.22 \pm 0.2$ | $82.10 \pm 0.1$ | $92.80 \pm 0.3$ | $92.17 \pm 0.1$ | $92.50 \pm 0.1$ | $\mathbf{93.20 \pm 0.1}$ |
| ImageNet-100 | $93.10 \pm 0.0$ | $94.20 \pm 0.0$ | $93.50 \pm 0.1$ | $93.60 \pm 0.1$ | $94.12 \pm 0.2$ | $93.25 \pm 0.1$ | $93.88 \pm 0.2$ | $\mathbf{94.95 \pm 0.1}$ |
| ImageNet-1K | $86.20 \pm 0.1$ | $87.30 \pm 0.0$ | $86.90 \pm 0.2$ | $86.80 \pm 0.1$ | $87.10 \pm 0.3$ | $86.95 \pm 0.1$ | $86.35 \pm 0.1$ | $\mathbf{88.20 \pm 0.2}$ |

Table 3: Calibration: REVAR is competitive with a host of strong baselines on the Expected Calibration Error metric (ECE).

| | Calibration Baselines | | | | | New Baselines | | REVAR |
| | CE | MMCE | Brier | FLSD-53 | AdaFocal | VR | MBR | Ours |
|---|---|---|---|---|---|---|---|---|
| DR(In-Dist.) | $7.7 \pm 0.1$ | $6.7 \pm 0.0$ | $5.8 \pm 0.1$ | $5.0 \pm 0.1$ | $\mathbf{3.6 \pm 0.1}$ | $7.4 \pm 0.1$ | $7.1 \pm 0.1$ | $3.8 \pm 0.1$ |
| DR(OOD) | $9.1 \pm 0.1$ | $7.9 \pm 0.1$ | $6.8 \pm 0.1$ | $6.1 \pm 0.1$ | $\mathbf{5.9 \pm 0.2}$ | $8.6 \pm 0.1$ | $8.4 \pm 0.3$ | $6.4 \pm 0.1$ |
| CIFAR-100 | $16.6 \pm 0.1$ | $15.3 \pm 0.1$ | $6.9 \pm 0.1$ | $5.9 \pm 0.1$ | $\mathbf{2.3 \pm 0.1}$ | $9.1 \pm 0.1$ | $10.7 \pm 0.1$ | $3.1 \pm 0.1$ |
| ImageNet-100 | $9.6 \pm 0.0$ | $9.1 \pm 0.0$ | $6.7 \pm 0.1$ | $5.8 \pm 0.1$ | $2.7 \pm 0.2$ | $8.2 \pm 0.1$ | $7.9 \pm 0.1$ | $\mathbf{2.7 \pm 0.1}$ |
| ImageNet-1K | $3.0 \pm 0.1$ | $9.0 \pm 0.0$ | $3.4 \pm 0.1$ | $16.1 \pm 0.1$ | $\mathbf{2.1 \pm 0.1}$ | $3.5 \pm 0.1$ | $3.2 \pm 0.1$ | $2.6 \pm 0.1$ |

**Scenario 4 - Dropping Features and covariate shift in validation set:**. We set $c = 1, G = 0, s > 0$ and only $X_o$ is available to the learner. In this case, the weights predicted by our meta-network roughly follows the relationship $\frac{\lambda_1}{W_l^T \Sigma(X_l | X_o) W_l} + \lambda_2 (x - \mu)^2$; this follows in a straightforward manner from the scenarios considered above. As before, from Sec. 4, when we increase the amount of covariate shift, the U-SCORE weights also reflect uncertainty from covariate shift more than that of label noise.

**Scenario 5 - Spurious Feature Shift:**. $c = 1, G = 0, s > 0$. Further $W_e = 0$. However, the learner sees $X_o$ alone for both test and validation. We now create a validation set using another distribution $\mathcal{N}(\mu', \sum')$ such that the distribution of $X_o$ remains same and the distribution of $X_l$ changes. This can be understood as a distribution shift setup where the core features required for predicting output for any instance remain the same but the background features change. In this case, the weights predicted by U-SCORE are close to uniform. This is because the model has to rely on core features $X_o$ alone, and there is no difference amongst training samples with respect to these features.

**Summary.** Tab. 1 summarizes the findings–our approach correlates strongly with theoretically desirable models for instance weights; further, when sources of uncertainty are mixed in different proportions, U-SCORE smoothly interpolates between them in determining instance weights (Sec. 4). Two additional key findings: the closest previous work (MWN (Shu et al., 2019), which proposed loss-based reweighting using a meta-network) performs significantly worse than our approach across scenarios. Interestingly, our own baseline (IBR, instance-based reweighting) improves across scenarios on MWN, but still falls significantly short of our full method. This provides strong evidence that *variance minimizing meta-regularization* is the key ingredient in the success of our approach.

## 5 EXPERIMENTS AND RESULTS

Having verified that REVAR accurately captures captures sources of uncertainty under various data generation process, we now evaluate it on a wide range of real-world scenarios and datasets. Since instance-level hardness or uncertainty is difficult to quantify in real-world settings, we use tasks such as selective classification or Neural Network calibration that evaluate uncertainty measures in aggregate form. We also show the general applicability of REVAR using experiments on the large-scale pretrained PLEX model (Tran et al., 2022) that show significant gains (appendix).

### 5.1 BASELINES.

For selective classification, we compare REVAR against several key baselines: Softmax-Response (SR) (Geifman & El-Yaniv, 2017), Monte-Carlo Dropout (MCD) (Gal & Ghahramani, 2016), Selec-

tiveNet (SN) (Geifman & El-Yaniv, 2019), Deep Gamblers (DG) (Liu et al., 2019) and Self-Adaptive Training (SAT) (Huang et al., 2020). Please refer to Sec. 2 for more information on these methods. We compare REVAR against recent proposals for calibration which show impressive results: Focal Loss (FLSD-53) (Mukhoti et al., 2020), MMCE (et al., 2018), Brier Loss (Brier et al., 1950) and AdaFocal (Ghosh et al., 2022) alongside the standard cross-entropy loss.

**Re-weighting Baselines:** We compare our method against other bi-level optimization based re-weighting baselines including Meta-Weight-Net (MWN) (Shu et al., 2019), Learning to Reweight (L2R) (Ren et al., 2018) and Fast Sample Re-weighting (FSR) (Zhang & Pfister, 2021), which have been designed explicitly label imbalance or random noise in labels setup, under various setups including selective classification (appendix), calibration (appendix) and input-dependent label uncertainty.

**New baselines:** We design two new baselines to separately measure bilevel optimization (reweighting) and meta-regularization: (a) ERM + Variance Reduction (VR) in training loss, and (b) Margin-based reweighting (MBR) of instances [2]. For both these baselines, we use softmax response for selection.

**Datasets.** We used the Diabetic Retinopathy (DR) detection dataset (kag, 2015), a significant real-world benchmark for selective classification, alongside the APTOS DR test dataset (Society, 2019) for covariate shift analysis. We also used CIFAR-100, ImageNet-100, and ImageNet-1K datasets. For the OOD test setting, we used Camelyon, WILDS, ImageNet-C,R,A. Furthermore, we utilize the Inst.CIFAR-100 (Xia et al., 2020), Clothing1M, IN-100H & CF-100H (Tran et al., 2022) datasets for input dependent noisy label settings. Please see appendix for dataset and preprocessing details.

## 5.2 PRIMARY REAL-WORLD SETTING: IN-DOMAIN VALIDATION, IN/OUT-DOMAIN TEST

### 5.2.1 REVAR OUTPERFORMS SOTA AT SELECTIVE CLASSIFICATION

Table 2 shows the results of the Area under the accuracy-rejection curve (Zhang et al., 2014) for REVAR and baselines on various datasets including Kaggle DR (in-dist.) & APTOS (OOD testing for model trained on Kaggle DR). Our method outperforms all other methods, showing its effectiveness as a measure of model uncertainty. In particular, we beat our own baselines VR,MBR that use variance reduction on *training loss*, and margin-based reweighting respectively on top of ERM. Accuracy & AUC at different coverage levels for all datasets are in the appendix.

**Scaling to large datasets:** Table2 shows that our method scales to large datasets such as Imagenet; we provide additional evidence (accuracy & AUC at various coverage levels) in the appendix.

We also compared against MCD and SAT on ImageNet-A/C/R benchmarks for robustness analysis. For all these experiments, the AUARC metric is provided in table 30.

### 5.2.2 REVAR IS COMPETITIVE AT CALIBRATION

Table 3 shows the results for this analysis for a pre-temperature-scaling setup. This is so that none of the approaches achieves any advantage of post-hoc processing and the evaluation is fair for all (see supplementary for more details). As can be observed, our results are competitive or better than SOTA for the calibration task. We also provide selective calibration (calibration at different coverages in selective classification), where we show larger gains over the baselines and demonstrate better calibration across the range–see supplementary materials.

Table 4: **AUARC:** In-Domain, OOD test set

| ImageNet-A | | | ImageNet-C | | | ImageNet-R | | |
|---|---|---|---|---|---|---|---|---|
| Ours | MCD | SAT | Ours | MCD | SAT | Ours | MCD | SAT |
| **9.98** | 8.44 | 8.91 | **65.9** | 63.7 | 64.2 | **68.8** | 66.8 | 67.1 |

Table 5: **AUARC:** OOD val, test set

| Data | MCD | SAT | Revar | Revar-PV |
|---|---|---|---|---|
| Camelyon | 74.99 | 75.16 | 76.32 | 78.12 |
| iWildCam | 76.07 | 76.17 | 77.98 | 79.86 |

Table 6: **Label Noise**: Re-weighting methods

| | MCD | MWN | L2R | FSR | Ours |
|---|---|---|---|---|---|
| Inst.CIFAR-100 | 61.12 | 65.89 | 67.12 | 70.21 | **71.87** |
| Clothing1M | 68.78 | 73.56 | 72.97 | 73.86 | **73.97** |

Table 7: **Label Noise**: Plex Model

| | Plex | Plex+ours |
|---|---|---|
| IN-100H | 0.75 | **0.71** |
| CF-100H | 0.49 | **0.47** |

[2]Since margin and loss are highly correlated, this is similar to loss-based reweighting

## 5.3 INPUT DEPENDENT LABEL NOISE

We now evaluate our methods on datasets comprising instance dependent label-noise. These include instance CIFAR-100 proposed in Xia et al. (2020), Clothing1M (Xiao et al., 2015) having noisy human labels and the label uncertainty setup proposed in PLEX (Tran et al., 2022) paper where instead of a single label, probabilities are assigned due to complex input and KL Divergence metric is used. On the CIFAR-100, Clothing datasets we compare with the other re-weighting methods designed for removing label noise using bi-level optimization including MWN, L2R, FSR (refer Sec. 5.1). In the Plex setup, we use our model on top of PLEX and analyze the improvements.

Tab. 33, Tab. 32 shows the results. Even though the re-weighting baselines have been designed for handling label noise, they are ineffective when this label noise is instance dependent, and are better suited for label imbalance/random flip (instance-independent). Our method handles these scenarios well, with signifciant gains. This matches the findings from the controlled study (Sec. 4).

## 5.4 SHIFTED VALIDATION SET

We now study the real-world scenario of shifted/OOD validation & test sets. We used theCamelyon, iWildCam datasets from the WILDS benchmark (Koh et al., 2021) where train, validation and test sets are each drawn from different domains. All methods are trained and tested on the same data splits. Table 31 compares REVAR and MCD, SAT on AUARC for selective classification. Again, we outperform the baselines, showing that REVAR efficiently handles models of uncertainty in domain shift settings; this reinforces the findings in our controlled scenarios 2 and 4 (Sec. 4).

**Using unlabeled test-domain samples.** We now consider another setup where the labeled validation set is in-domain, but we can use unlabelled OOD samples from the test domain. Since our $l_{eps}$ regularizer does not use labels, we propose using it for unsupervised domain adaptation on these samples. REVAR-PV pools in-domain and (unlabeled) test-domain meta-losses, while REVAR-DSV only uses test-domain samples for the meta-loss. This also corresponds to scenario 2 (Sec. 4) since the main component in determining hardness (variance minimization) is applied on OOD examples. REVAR-PV handily beats other approaches in this setting (Tab. 8), suggesting that both generalization and variance minimization are important. This aligns with Sec. 4: as we increase covariate shift in validation, the hardness coefficient $\lambda_2$ dominates in determining U-SCORE scores. Tab. 31 provides further evidence, where validation is OOD and shifted approximately towards test data due to weak supervision from unlabelled instances. See appendix for results on ImageNet-C,R,A datasets.

Table 8: Comparing REVAR variants for unsupervised domain adaptation. REVAR-PV pools in-domain and out-of-domain validation data, while REVAR-DSV only uses domain-shift validation data in the meta-objective.

| Coverage | REVAR | REVAR-PV | REVAR-DSV | VR-DSV | VR-PV |
|---|---|---|---|---|---|
| 1.0 | 86.1 | **88.3** | 85.3 | 87.4 | 87.2 |
| 0.8 | 88.1 | **90.6** | 87.4 | 88.6 | 88.8 |
| 0.6 | 89.9 | **91.7** | 88.2 | 89.9 | 89.6 |
| 0.4 | 91.4 | **93.1** | 88.9 | 91.9 | 91.8 |

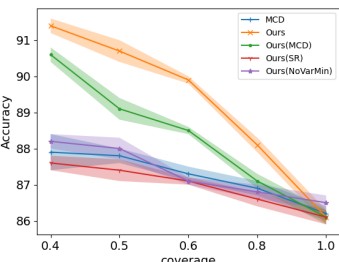

Figure 2: Lesion Study (DR)

## 5.5 REVAR LESION ANALYSIS

We examined the contribution of the various components of our proposal to the overall efficacy of REVAR in selective classification. We study the following variants: (1) REVAR-**NoVarMin:** Drops the variance-reduction meta-regularization, (2) REVAR-**SR** and REVAR-**MCD**: Uses REVAR classifier's logits or MCD respectively at test time instead of U-SCORE, (3) **MCD:** baseline.

Figure 2 shows this comparison on the DR dataset under country shift, for the accuracy metric. We make the following observations: (1) REVAR performs best, and dropout variance reduction plays a very large role. (2) REVAR-MCD beats MCD: REVAR classifiers are inherently more robust. (3) REVAR beats REVAR-SR and REVAR-MCD (differing only in test-time scoring): U-SCORE is a better measure of uncertainty than MCD or logits, even on the more-robust REVAR classifier.

## REPRODUCIBILITY STATEMENT

We use the ResNet-50 architecture as classifier for the diabetic retinopathy experiments. Each experiment has been run 5 times and the associated mean and standard deviation are reported. For MC-Dropout baseline and our meta-objective, we calculate the uncertainty by 10 forward passes through the classifier. We used a learning rate of 0.003 and a batch size of 64 to train each model in each of the experiments. For our re-weighting scheme, we separate 10 percent of the data as the validation set. For CiFAR-100, ImageNet-1k, Clothing1M also we have used ResNet-50 and for ImageNet-100 we use VGG-16 in all experiments, inspired by recent work (Huang et al., 2020; Liu et al., 2019). A batch size of 128 is used and an initial learning rate of $1e - 2$ with a momentum of 0.9 is used. For U-SCORE we have used a learning rate of $1e - 4$ with a momentum of 0.9 and batch size same as classifier for all the experiments. For clothing dataset, we have used the same processing and hyper-parameter setup as MWN (Shu et al., 2019). For efficiency and better training, we update the U-SCORE for every $K = 15$ steps of the classifier. Also, we warm start the classifier by training it without the U-SCORE for first 25 epochs. A weight decay of $10^{-4}$ is used for both the networks. For all experiments, training is done for 300 epochs. For the unlabelled test instances from Kaggle data to APTOS data or ImageNet to ImageNet-A,R,C data we split the training into 230 epochs without unlabelled images and 70 epochs combining the unlabelled images along with the source data for a total of 300 epochs. Around 10% of the test images are sampled as this unlabelled setup for this setup. For the PLEX model experiments, we just apply our technique on top its existing implementation, keeping same hyper-parameters and using a learning rate of $1e - 3$ for the U-SCORE.

For synthetic data, $W_{data}, G_{data}$ are matrices with each of their elements sampled from $\mathcal{N}(5, 10)$, $\mathcal{N}(12, 18)$ respectively. The $\mu, \sum$ for $X_{train}$ are generated by sampling 72 values from $\mathcal{N}(1, 10)$, $\mathcal{N}(5, 10)$ respectively. For scenarios 2, 4 the values of $s$ are kept at 25, 50 respectively.

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

# Appendix

## A ReVaR Algorithm

---

**Algorithm 1** ReVaR training procedure.

---

**Require:** Prediction Network parameters $\theta$, U-Score parameters $\Theta$, learning rates $(\beta_1, \beta_2)$, dropout rate $p_{drop}$, training data $\{x_i, y_i\}_{i=1}^N$, validation data $\{x_i^s, y_i^s\}_{i=1}^M$, U-Score update interval $\mathcal{M}$.

**Ensure:** Robustly trained classifier parameters $\theta^*$, U-Score parameters $\Theta^*$ to predict uncertainty.

1: Randomly initialize $\theta$ and $\Theta$, $t = 1$;
2: **for** e = 1 **to** E **do**                                           ▷ E: number of epochs
3:     sample a minibatch $\{(x_i, y_i)\}_{i=1}^n$ from training data;          ▷ n denotes the batch size
4:     **if** $t\%\mathcal{M} == 0$ **then**
5:         Create a copy of the current prediction model, denoting parameters by $\hat{\theta}$
6:         sample minibatch $\{(x_i^v, y_i^v)\}_{i=1}^m$ from validation data
7:         $\hat{\theta} \leftarrow \hat{\theta} - \beta_1 \nabla_{\hat{\theta}} \sum \left(l(f_{\hat{\theta}}(x), y)\right)$          ▷ Update the copy of prediction model
8:         $\Theta \leftarrow \Theta - \beta_2 \nabla_{\Theta} \sum \left(l(y_i^v, f_{\hat{\theta}}(x_i^v) + l_{eps}(\hat{\theta}, x_i^v)\right)$      ▷ Update U-Score using Eq. 5
9:     **end if**
10:     $\theta \leftarrow \theta - \beta_1 \nabla_{\theta} \sum g_{\Theta}(x_i) l(f_{\theta}(x_i), y_i)$;              ▷ Update the prediction model
11:     $\theta^* \leftarrow \theta$; $\Theta^* \leftarrow \Theta$; $t \leftarrow t + 1$
12: **end for**

---

## B Updates for the bilevel optimization

Revisiting the bi-level optimization objective proposed in the paper:

$$\theta^* = \arg\min_{\theta} \frac{1}{N} \sum_{i=1}^{N} g_{\Theta}(x_i) \cdot l(y_i, f_{\theta}(x_i))$$
$$s.t. \ \Theta^* = \arg\min_{\Theta} \mathcal{L}_{meta}(X^s, Y^s, \theta^*) \tag{8}$$

where $\theta, \Theta$ correspond to model parameters for the primary & U-Score models ($f_{\theta}$ and $g_{\Theta}$ respectively), $(x_t, y_t)$ denote the input-output pair corresponding to the training set and $l$ is the cross-entropy cost function for the classifier. As discussed in the paper, the loss $\mathcal{L}_{meta}(X^s, Y^s, \theta^*)$ is as follows:

$$\mathcal{L}_{meta}(X^s, Y^s, \theta^*) = \frac{1}{M} \sum_{j=1}^{M} \left(\mathcal{L}_{eps}(x_j^s, \theta^*) + l(y_j, f_{\theta}^*(x_j^s))\right) \tag{9}$$

where $x_j^s, y_j^s$ are input instance and its corresponding output belonging to the specialized set. This formulation results in a nested optimization which involves updating U-Score$(\Theta)$ at the outer level using the cross entropy loss and variance of the classifier parameters $\theta^*$, generated by sampling different dropout masks The backpropagation based update equation for $\Theta$ at epoch $t$ ($\Theta_t$) is as follows:

$$\Theta_{t+1} = \Theta_t - \frac{\alpha}{M} \sum_{j=1}^{M} \nabla_{\Theta} \left(\mathcal{L}_{eps}(x_j^s, \theta^*) + l(y_j, f_{\theta_t^*}(x_j^s))\right) \tag{10}$$

where $\alpha$ is the step size. The gradient term in the above equation can be further simplified to:

$$-\frac{\alpha}{M} \cdot \sum_{j=1}^{M} \nabla_{\theta}^* \left(\mathcal{L}_{eps}(x_j^s, \theta^*) + l(f_{\theta^*}(x_j^s), y_j^s)\right)\Bigg|_{\theta_t^*} \nabla_{\Theta}(\theta^*)\Bigg|_{\Theta_t} \tag{11}$$

where $\nabla_{\theta}(.)\big|_{\theta_t}$ denotes evaluating the gradient at $\theta = \theta_t$. Solving this optimization is quite a time-consuming process since it requires implicit gradient $\frac{\partial \theta^*}{\partial \Theta}$ and also completely optimizing the

inner loop for one step in outer loop. Thus, we also follow the approximations used in Shu et al. (2019) and convert this nested to an alternating optimization setup for $\Theta$ and $\theta$. Thus, now $\theta^*$ in the above equation can be replaced with $\theta$. To implement this, we again follow MWN(Shu et al., 2019) and update $\Theta$ by using a copy of the $\theta$ $i.e.$, $\hat{\theta}$ at every instant when U-SCORE is updated. This makes the optimization process easy to interpret as well as stable. At any instant $t+1$, it involves first calculating $\hat{\theta}$ using the following eq.:

$$
\hat{\theta} = \theta_t - \frac{\beta}{N} \cdot \sum_{i=1}^{N} \nabla_\theta \left( g_\Theta(x_i) \cdot l(y_i, f_\theta(x_i)) \right) \Bigg|_{\theta_t, \Theta_t}
\tag{12}
$$

where $\beta$ is the step size. Now, differentiating this w.r.t. $\Theta$:

$$
\nabla_\Theta(\hat{\theta}) = -\frac{\beta}{N} \cdot \sum_{i=1}^{N} \nabla_\Theta g_\Theta(x_i) \Bigg|_{\Theta_{t+1}} \cdot \nabla_\theta l(y_i, f_\theta(x_i)) \Bigg|_{\theta_t}
\tag{13}
$$

In eq. 11, $\theta^*$ is replaced by $\hat{\theta}$ and the last term $\frac{\partial \hat{\theta}}{\partial \Theta}$ can be replaced by this last equation which will modify the equation 11 to:

$$
\frac{\alpha\beta}{MN} \cdot \sum_{j=1}^{M} \nabla_{\hat{\theta} \left( \mathcal{L}_{eps}(x_j^s, \hat{\theta}) + l(f_{\hat{\theta}}(x_j^s), y_j^s) \right)} \Bigg|_{\hat{\theta}_t} \sum_{i=1}^{N} \nabla_\Theta g_\Theta(x_i) \Bigg|_{\Theta_{t+1}} \cdot \nabla_\theta l(y_i, f_\theta(x_i)) \Bigg|_{\theta_t}
\tag{14}
$$

Rearranging terms:

$$
\frac{\alpha\beta}{MN} \cdot \sum_{i=1}^{N} \nabla_\Theta g_\Theta(x_i) \cdot \sum_{j=1}^{M} \nabla_\theta \left( \mathcal{L}_{eps}(x_j^s, \theta) + l(f_\theta(x_j^s), y_j^s) \right) \cdot \nabla_\theta l(y_i, f_\theta(x_i)) \Bigg|_{\theta_t, \Theta_{t+1}}
\tag{15}
$$

We now write the update equation for classifier parameters $\theta$ at time $t+1$ involving the re-weighting network parameters $\Theta_{t+1}$:

$$
\theta_{t+1} = \theta_t - \frac{\beta}{N} \cdot \sum_{i=1}^{N} \nabla_\theta \left( g_\Theta(x_i) \cdot l(y_i, f_\theta(x_i)) \right) \Bigg|_{\theta_t, \Theta_{t+1}}
\tag{16}
$$

The equation can be further simplified since $\theta$ is not dependent on $\Theta$:

$$
\theta_{t+1} = \theta_t - \frac{\beta}{N} \cdot \sum_{i=1}^{N} \left( g_\Theta(x_i) \cdot \nabla_\theta l(y_i, f_\theta(x_i)) \right) \Bigg|_{\theta_t, \Theta_{t+1}}
\tag{17}
$$

This completes the derivation for update equation. Given the bilevel optimization formulation, we choose to update $\Theta$ at every $K$ updates of $\theta$ based on the assumption that these $K$ updates of $\theta$ can be used to approximate $\theta^*$.

## C  IMPROVING PLEX MODEL

We now evaluate our proposed method applied to a large pretrained model, specifically the recently proposed PLEX (Tran et al., 2022) model, pretrained on large amounts of data. Tran et al. (2022) show that the rich learned representations in PLEX yield highly reliable predictions and impressive performance on various uncertainty related benchmarks like selective classification, calibration, label uncertainty etc. These applications all require fine-tuning on target data; for our version of PLEX, we replaced their standard unweighted fine-tuning with a weighted fine-tuning combined with the U-SCORE and our associated meta objective.

**Datasets and Tasks.** In addition to selective classification and calibration, the PLEX paper studies a *label uncertainty task* which requires estimating the KL Divergence between the predicted and actual label distribution. For Selective Classification, we compare accuracies at various coverage labels on the DR dataset with covariate shift test set. For calibration, we use the in-distribution and OOD

datasets used in the PLEX paper and also compare with approaches like Focal loss, MMCE on these datasets. Finally, for the label uncertainty task, we use the ImageNet-100H and CIFAR-100H datasets used in the PLEX paper.

REVAR **improves PLEX.** Table 9a shows Expected Calibration Error (ECE) across datasets; REVAR improves PLEX ECE by significant margins in both In-Distribution (upto around 12%) and Out-of-Distribution (upto around 13%) datasets. Table 9b shows the result of label uncertainty experiment on the ImageNet-100H and CIFAR-100H datasets, showing KL Divergence between the available and predicted probability distribution. Again, using our approach on top of PLEX yields upto 4% gains. Table 10 shows a similar trend in selective classification where we improve PLEX performance at most coverages, and also at 100% coverage, *i.e.*, complete data. This showcases the effectiveness of REVAR at capturing the entire range of uncertainty.

Taken together, these results show the potential value of REVAR in easily providing gains on top of large, powerful pretrained models, particularly when such *foundation models* are becoming increasingly common.

Table 9: Analysis of Plex model combined with ours for calibration and label uncertainty tasks.

| (a) Calibration (ECE) | | | | | | (b) Label Uncertainty | | |
|---|---|---|---|---|---|---|---|---|
| | FSLD-53 | MMCE | Ours | Plex | Plex+ours | | Plex | Plex+ours |
| In-Dist. | 6.80 | 7.30 | 5.10 | 0.93 | **0.81** | IN-100H | 0.75 | **0.71** |
| OOD | 13.80 | 14.50 | 12.20 | 7.50 | **6.20** | CF-100H | 0.49 | **0.47** |

Table 10: Selective Classification on DR dataset (both in-distribution and country-shift).

| Dataset | Kaggle (*in-distribution*) | | | | | | APTOS (*out-of-distribution*) | | | | | |
|---|---|---|---|---|---|---|---|---|---|---|---|---|
| Coverage | 0.2 | 0.4 | 0.6 | 0.8 | 1.0 | AUC | 0.2 | 0.4 | 0.6 | 0.8 | 1.0 | AUC |
| Plex | 97.15 | 96.09 | 95.37 | 93.89 | 89.87 | 96.30 | 92.47 | 87.12 | 88.46 | 90.37 | 88.13 | 90.40 |
| Plex+ours | **97.65** | **96.88** | **95.67** | **94.66** | **90.34** | **96.98** | **92.96** | **88.16** | **89.02** | **90.91** | **88.86** | **90.95** |

# D    EXPERIMENTAL DETAILS AND RELATED ANALYSIS

## D.1    DATASETS AND METRICS

We now discuss various datasets we have used to evaluate our method for different tasks.

Table 11: Dataset statistics

| Dataset | #labels | Train | Val | Test | Size |
|---|---|---|---|---|---|
| DR (Kaggle) | 5 (binarized) | 35,697 | 2617 | 42,690 | $512 \times 512$ |
| DR (APTOS) | 5 (binarized) | - | - | 2917 | $512 \times 512$ |
| ImageNet-100 | 100 | 130,000 | - | 5000 | $256 \times 256$ |
| CIFAR-100 | 100 | 50000 | - | 10000 | $28 \times 28$ |

**Diabetic Retinopathy.** The recently proposed Kaggle Diabetic Retinopathy (DR) Detection Challenge (kag, 2015) dataset and the APTOS dataset (Society, 2019) are used as an uncertainty benchmark for Bayesian learning (Filos et al., 2019). The Kaggle dataset consists of 35,697 training images, 2617 validation and 42,690 testing images, whereas the APTOS dataset consists of 2917 evaluation images for testing ddomain generalization for DR detection. In particular, the APTOS dataset is collected from labs in India using different equipment, and is distributionally different from the Kaggle dataset. Both datasets label input retina images with the severity of diabetic retinopathy at 5 grades– 0-4 as No DR, mild, moderate, severe and proliferative. Similar to (Filos et al., 2019), we formulate a binary classification problem grouping grades 0-2 as negative class and 3,4 as positive class. We

focus primarily on this dataset given its real-world value and role as a benchmark specifically relevant to selective classification (Filos et al., 2019).

**ImageNet-1K and Shifts**. This dataset comprises of around 1.4M images with around 1.28M for training, 50k for validation and 100k testing images. It invovles solving a classification problem with 1000 classes. Furthermore, we also include results on popular shifted versions of ImageNet: **ImageNet-A** comprising hard examples misclassified by Resnet-50 on ImageNet, **ImageNet-C** comprising 15 different kinds of noises at various severity levels simulating a practical scenario and **ImageNet-R** comprising changes in style/locations, blurred images or various other changes common in real-world.

**Other image datasets**. We study other image classification datasets commonly used for evaluating selective classification methods. including the **CIFAR-100** dataset (Krizhevsky et al., 2009) (10 categories of natural images). We also evaluate on a subset of the widely used ImageNet dataset–**ImageNet-100** (Tian et al., 2020)–consisting of 100 randomly selected classes from the 1k classes in the original ImageNet dataset. This serves as a larger-scale stress-test of selective classification given the relatively larger image size, dataset size, and number & complexity of categories. Alongside these, for input dependent label noise Inst.CIFAR-100 with $\epsilon = 0.2$ (Xia et al., 2020) PLEX label uncertainty datasets (IN-100H, CF-100H) (Tran et al., 2022) and Clothing1M (Xiao et al., 2015) datasets are used.

## D.2    BASELINES.

Below we briefly discuss the exhaustive list of baselines used in the paper.
**MCD** (Gal & Ghahramani, 2016). It applies dropout to any neural network and take multiple passes during inference and calculates the entropy of the averaged soft-logits for uncertainty.
**DG** (Liu et al., 2019). It updates the training objective by adding a background class and t the inference time abstains from prediction if the probabity of instance being in that class is higher than some threshold.
**SN** (Geifman & El-Yaniv, 2019). It proposed using an auxillary network to predict a confidence score whether model wants to predict for an instance at a pre-defined coverage rate.
**SAT** (Huang et al., 2020). It uses a target label as exponential moving average of model predictions and label throughout the training and uses an extra class as the selection function, using an updated objective to enforce uncertain examples into the extra class.
**Brier Loss**. Squared error betwee softmax predicted logits and the ground truth label vector. (Brier et al., 1950). Mea
**FLSD-53** (Mukhoti et al., 2020). It uses focal loss and proposes a method for selecting the appropriate hyper-parameter for it.
**Adafocal** (Ghosh et al., 2022). It updates the hyperparameter of the focal loss independently for each instance based on its value at previous step, utilizing validation feedback.
**MWN** (Shu et al., 2019). It uses a bi-level optimization setup comprising a meta-network which takes loss as input and predicts weights for train instances such that validation performance is maximized (Shu et al., 2019).
**L2R** (Ren et al., 2018). It uses one free-parameter per training instance as the weight of its loss while updating model. These free parameters are learned using meta-learning to optimize validation performance.
**FSR** (Zhang & Pfister, 2021). Similar to L2R excpet that it doesn't require a pre-defined clean validation set and at fixed intervals keep interchanging train and val examples based on how much updating on any instance is changing the validation set loss.

**Metrics.** For selective classification, we measure and report accuracy for all datasets. In addition, for the DR dataset, we also measure AUC, a measure of performance that is robust to class imbalance, and to potential class imbalance under selective classification. The accuracy and AUC are measured on those data points selected by each method for prediction, at the specified target coverage. We also measure *selective calibration*, i.e., calibration error (ECE (Naeini et al., 2015)) measured on only the data points selected by the method for the specified coverage. All metrics reported are averages $\pm$ standard deviation over 5 runs of the method with different random initializations.

Table 11 summarizes the various datasets used in our experiments, and their characteristics.

### D.3 TRAINING & EVALUATION DETAILS

We use the ResNet-50 architecture as classifier for the diabetic retinopathy experiments. Each experiment has been run 5 times and the associated mean and standard deviation are reported. For MC-Dropout baseline and our meta-objective, we calculate the uncertainty by 10 forward passes through the classifier. We used a learning rate of 0.003 and a batch size of 64 to train each model in each of the experiments. For our re-weighting scheme, we separate 10 percent of the data as the validation set. For CiFAR-100 also we have used ResNet-50 and for ImageNet-100 we use VGG-16 in all experiments, inspired by recent work (Huang et al., 2020; Liu et al., 2019). A batch size of 128 is used and an initial learning rate of $1e-2$ with a momentum of 0.9 is used. For U-SCORE we have used a learning rate of $1e-4$ with a momentum of 0.9 and batch size same as classifier for all the experiments. For efficiency and better training, we update the U-SCORE for every $K = 15$ steps of the classifier. Also, we warm start the classifier by training it without the U-SCORE for first 25 epochs. A weight decay of $10^{-4}$ is used for both the networks.

For all experiments, training is done for 300 epochs. For the unsupervised domain adaptation from Kaggle data to APTOS data, we split the training into 230 epochs without unlabelled images and 70 epochs combining the unlabelled images along with the source data for a total of 300 epochs.

## E  DETAILED RESULTS ON SELECTIVE CLASSIFICATION

### E.1  DIABETIC RETINOPATHY DATASET

We present our main results on a large real-world application of selective classification: Diabetic retinopathy detection (kag, 2015). Our evaluation considers both in-distribution data, as well as a separate test set from a different geographic region collected using different equipment– this is an evaluation of *test-time generalization* under domain shift, without any additional learning.

Figure 3 shows a comparison of all methods on selective classification for the Kaggle DR dataset (first row) alongwith domain generalization results (Country shift evaluated using APTOS dataset, second row)). The columns present different metrics for each task: AUC (column 1), accuracy (column 2), and *selective calibration* error (column 3). We see that REVAR consistently outperforms the other methods on both tasks and all metrics. In particular, the robust gains on AUC (column 1, upto 1.5% absolute, see Table 12) for both in-distribution and domain shift tasks are compelling. Note that although the results are reflected in accuracy metrics as well (column 2, upto 2% absolute gains on the domain shift task, see Table 12), AUC is less susceptible to class imbalances and therefore a more reliable metric. Also, column 3 shows robust improvement in calibration on both in-domain and out-of-domain data (ECE metric, lower is better), suggesting that the U-SCORE indeed better represents classifier uncertainty, and thereby improves on selective classification. Finally, we note that the improvement in calibration, a widely used metric of classifiers' ability to capture and represent uncertainty, suggests that REVAR may have broad applications beyond selective classification (see e.g., (Tran et al., 2022)).

A note of interest is that AUC for all methods *reduces* in the domain shift task as the selectivity is increased. This is the opposite of expected behavior, where accuracy and AUC should generally increase as the classifier becomes more selective. The data suggests a significant change in the two data distributions that appears to partially *invert* the ranking order–i.e., all classifiers appear to be more accurate for instances they are less confident about. The robust gains of REVAR suggest that it is less susceptible to such drastic shifts in distribution.

### E.2  IMAGENET-100

We replicated our findings on other datasets commonly used for studying selective classification in the literature. This includes Imagenet-100 (Table 13). REVAR retains an edge over the other baselines in each of these datasets. In particular, the Imagenet-100 dataset is sufficiently complex, given the much larger larger number of classes (100) on a substantial training and evaluation set of higher-resolution images. REVAR's superior performance on this dataset shows its potential for scaling to harder selective classification problems.

Table 12: Comparison on the Kaggle dataset and the APTOS dataset under the country shift setup.

| | Kaggle Dataset (*in-distribution*) | | | | | | | | | |
|---|---|---|---|---|---|---|---|---|---|---|
| | 40% retained | | 50% retained | | 60% retained | | 80% retained | | 100% retained | |
| | AUC(%) | Acc.(%) | AUC(%) | Acc.(%) | AUC(%) | Acc.(%) | AUC(%) | Acc.(%) | AUC(%) | Acc.(%) |
| MCD | $96.3 \pm 0.1$ | $97.8 \pm 0.0$ | $95.2 \pm 0.1$ | $97.1 \pm 0.1$ | $93.7 \pm 0.3$ | $95.3 \pm 0.2$ | $92.3 \pm 0.2$ | $92.8 \pm 0.1$ | $91.2 \pm 0.2$ | $90.6 \pm 0.1$ |
| DG | $95.9 \pm 0.1$ | $97.2 \pm 0.1$ | $94.4 \pm 0.2$ | $96.4 \pm 0.1$ | $93.3 \pm 0.2$ | $95.1 \pm 0.1$ | $92.5 \pm 0.3$ | $93.1 \pm 0.1$ | $91.3 \pm 0.3$ | $90.8 \pm 0.1$ |
| SN | $95.8 \pm 0.1$ | $97.0 \pm 0.1$ | $94.2 \pm 0.2$ | $96.1 \pm 0.1$ | $93.5 \pm 0.3$ | $95.2 \pm 0.1$ | $92.8 \pm 0.1$ | $93.4 \pm 0.1$ | $91.4 \pm 0.3$ | $90.9 \pm 0.2$ |
| SAT | $96.5 \pm 0.0$ | $97.9 \pm 0.0$ | $95.0 \pm 0.1$ | $96.8 \pm 0.1$ | $93.9 \pm 0.2$ | $\mathbf{95.6 \pm 0.1}$ | $92.7 \pm 0.2$ | $93.6 \pm 0.2$ | $\mathbf{91.7 \pm 0.3}$ | $\mathbf{91.1 \pm 0.2}$ |
| Ours | $\mathbf{97.5 \pm 0.1}$ | $\mathbf{98.4 \pm 0.0}$ | $\mathbf{96.3 \pm 0.2}$ | $\mathbf{97.4 \pm 0.1}$ | $\mathbf{94.4 \pm 0.3}$ | $95.5 \pm 0.2$ | $\mathbf{92.9 \pm 0.2}$ | $\mathbf{93.8 \pm 0.2}$ | $91.5 \pm 0.3$ | $91.0 \pm 0.1$ |
| | APTOS Dataset (*country shift*) | | | | | | | | | |
| MCD | $79.8 \pm 0.8$ | $87.9 \pm 0.5$ | $87.2 \pm 0.4$ | $87.8 \pm 0.2$ | $89.1 \pm 0.2$ | $87.3 \pm 0.2$ | $91.4 \pm 0.3$ | $86.9 \pm 0.2$ | $93.6 \pm 0.3$ | $86.2 \pm 0.2$ |
| DG | $83.7 \pm 0.6$ | $87.5 \pm 0.3$ | $88.1 \pm 0.3$ | $87.1 \pm 0.2$ | $90.1 \pm 0.6$ | $86.9 \pm 0.2$ | $91.9 \pm 0.2$ | $86.2 \pm 0.2$ | $\mathbf{93.7 \pm 0.6}$ | $86.1 \pm 0.1$ |
| SN | $86.2 \pm 0.4$ | $88.4 \pm 0.4$ | $88.1 \pm 0.2$ | $88.3 \pm 0.2$ | $89.7 \pm 0.3$ | $87.5 \pm 0.1$ | $91.1 \pm 0.2$ | $87.2 \pm 0.1$ | $93.2 \pm 0.2$ | $86.3 \pm 0.1$ |
| SAT | $87.3 \pm 0.3$ | $89.8 \pm 0.3$ | $88.7 \pm 0.2$ | $89.2 \pm 0.2$ | $89.3 \pm 0.2$ | $87.9 \pm 0.1$ | $91.3 \pm 0.3$ | $87.1 \pm 0.2$ | $92.7 \pm 0.3$ | $\mathbf{86.9 \pm 0.2}$ |
| Ours | $\mathbf{89.2 \pm 0.4}$ | $\mathbf{91.4 \pm 0.2}$ | $\mathbf{90.2 \pm 0.3}$ | $\mathbf{90.7 \pm 0.3}$ | $\mathbf{90.9 \pm 0.2}$ | $\mathbf{89.9 \pm 0.1}$ | $\mathbf{91.8 \pm 0.2}$ | $\mathbf{88.1 \pm 0.2}$ | $92.3 \pm 0.3$ | $86.1 \pm 0.2$ |

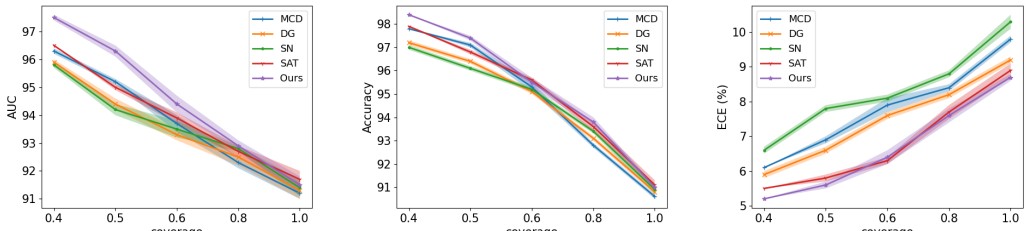

(a) DR dataset (in-distribution AUC)  (b) DR dataset (in-distribution Accuracy)  (c) DR dataset (in-distribution calibration)

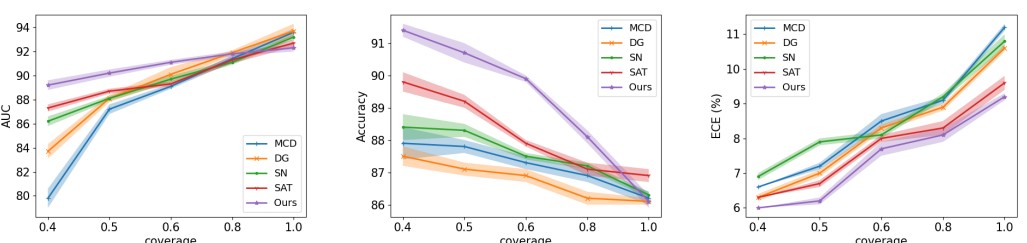

(d) DR dataset (country shift AUC)  (e) DR dataset (country shift Accuracy)  (f) DR dataset (country shift calibration)

Figure 3: Selective classification results on diabetic retinopathy dataset. REVAR shows robust improvement in AUC in both in-domain and domain-shift scenarios (panels (a,b)). Accuracy measures also show similar trends, with large improvements in domain shift conditions (panels (c,d)). Finally, selective calibration error measures (calibration of selected data points, panels (e,f)) show that better calibration is a key underlying factor for REVAR's performance. See text for details.

In all datasets we see a pattern of increasing gap as the coverage is reduced, suggesting that REVAR is able to identify and retain the highest-confidence test instances better than the other methods.

Table 13: Comparison on the ImageNet-100 dataset.

| | 0.6 | 0.7 | 0.8 | 0.9 | 1 |
|---|---|---|---|---|---|
| MCD | $2.55 \pm 0.4$ | $3.62 \pm 0.3$ | $6.34 \pm 0.3$ | $9.34 \pm 0.4$ | $13.74 \pm 0.3$ |
| DG | $2.31 \pm 0.4$ | $3.41 \pm 0.3$ | $5.36 \pm 0.4$ | $\mathbf{8.58 \pm 0.4}$ | $13.62 \pm 0.5$ |
| SN | $2.13 \pm 0.2$ | $3.51 \pm 0.2$ | $6.07 \pm 0.1$ | $9.56 \pm 0.2$ | $13.88 \pm 0.2$ |
| SAT | $1.89 \pm 0.2$ | $2.86 \pm 0.3$ | $5.38 \pm 0.2$ | $8.89 \pm 0.3$ | $\mathbf{13.70 \pm 0.4}$ |
| Ours | $\mathbf{1.48 \pm 0.2}$ | $\mathbf{2.32 \pm 0.4}$ | $\mathbf{5.08 \pm 0.3}$ | $8.67 \pm 0.2$ | $13.73 \pm 0.3$ |

E.3    IMAGENET-1K

For a large scale demonstration of our approach, we now present the results on the ImagaNet-1K dataset for the selective classification setup. We utilize the complete 1.28M train images to update the classifier and use the 50k validation images to update the U-SCORE. Table 14 shows the results for this evaluation against the existing baselines. It contains analysis on five different coverages ranging from 0.4 to 1.0. It can be observed that our method is the best-performing at lower coverage levels (0.4,0.5) and also at moderately high coverage levels (0.8). Also, it is able to provide gains upto 1.5% in accuracy (0.5) and shows a significant gain of 0.88% over all the existing baselines at the coverage level of 0.4.

Table 14:  Comparison on the ImageNet-1k dataset

|       | 0.4   | 0.5   | 0.6   | 0.8   | 1.0   |
|-------|-------|-------|-------|-------|-------|
| SAT   | 95.34 | 90.12 | 87.16 | 82.12 | 75.31 |
| DG    | 95.27 | 90.53 | 87.27 | 82.06 | **75.44** |
| SN    | 95.19 | 90.22 | **87.74** | 81.78 | 75.02 |
| Ours  | **96.22** | **91.67** | 87.64 | **83.38** | 75.21 |

Table 15: U-SCORE architecture ablation.

|       | 0.4  | 0.5  | 0.6  | 0.8  | 1.0  |
|-------|------|------|------|------|------|
| RN-18 | 91.4 | 90.7 | 89.9 | 88.1 | 86.1 |
| RN-32 | 91.3 | 90.5 | 90.1 | 88.2 | 85.9 |
| RN-50 | 91.1 | 90.4 | 90.2 | 88.3 | 86.0 |

E.4    FURTHER ANALYZING THE UNLABELLED TEST DOMAIN INSTANCES SCENARIO

We further test the importance of utilizing unlabelled examples from test domain, given the in-domain validation set setting, in our REVAR-PV variant, which has proven to be better at capturing uncertainty than REVARin the experiments provided in main paper. We further verify this by testing on the ImageNet-A,R,C datasets by using ImageNet as the training, in-domain val set. Each of them inherits a significant shift from ImageNet. The results are provided in table 16. it can be observed that again REVAR-PV comes out to be significantly better than REVARin terms of modelling the uncertainty for this complete generative setup.

Table 16: Comparison of REVARwith in-domain validation set and its variant utilizing unlablled instances from test domains REVAR-PV on ImageNet-A,R,C datasets inheriting domain shift.

| Method   | ImagNet-A | ImageNet-C | ImageNet-R |
|----------|-----------|------------|------------|
| Revar    | 9.98      | 65.9       | 68.8       |
| Revar-PV | 12.6      | 69.0       | 70.7       |

F    ARCHITECTURE, MODEL SIZES, COST

F.1    U-SCORE ARCHITECTURES

For all the experiments discussed till now, we have used a ResNet-18 (Pretrained) as the U-SCORE for all the experiments. We now perform an ablation on the choice of the U-SCORE architectures including ResNet-18, ResNet-32 and ResNet-50 for the DR dataset (OOD), in table 15. Given the limited data available to train the meta-network, big architecture might be sub-optimal, verified by the results in the table. However, for large-scale datasets like ImageNet-1K, increasing capacity can be more helpful at the cost of increased computation. All these experiments use a ResNet-50 as the classifier architecture. We also analyze a different architecture for the classifier (WRN-28-10) in the appendix.

**Vision Transformers**. Inspired by the recent success of Vision Transformer Models, we also analyze this architecture for the U-SCOREŠpecifically, we test a ViT-small based U-SCORE against the ResNet-101 based U-SCORE on the DR dataset under country shift setup, both having similar number of parameters (45M, 48M respectively). We also do a similar comparison on the ImageNet-1k dataset for further assurance. The classifier architecture is same as the U-SCORE architecture. Table 17 provides the analysis for this experiment.

Table 17: Comapring ViT-S architecture for both U-SCORE and classifier against the RN-101 architecture for both.

| | DR (OOD) | | | | | ImageNet-1k | | | | |
|---|---|---|---|---|---|---|---|---|---|---|
| | 0.4 | 0.5 | 0.6 | 0.8 | 1.0 | 0.4 | 0.5 | 0.6 | 0.8 | 1.0 |
| RN-101 | 91.1 | 90.4 | 90.2 | 88.3 | 86.0 | 96.6 | 92.3 | 88.4 | 84.4 | 76.1 |
| ViT-S | 92.7 | 91.6 | 91.1 | 89.3 | 87.2 | 96.9 | 93.2 | 89.9 | 86.3 | 78.2 |

## F.2 CHANGING MODEL ARCHITECTURES

We examine the effect of backbone in evaluation of our proposed scheme. Specifically, we compare the top-2 performing baselines namely SelectiveNet (SN) and Self-Adaptive Training (SAT) with our method using a Wide-ResNet-28-10 backbone with around 1.5 times parameters compared to the ResNet-50 backbone used in the paper along with a different architecture. We do this for the Diabetic Retinopathy data as well as the Imagenet-100 data. Table 18 shows the analysis for diabetic retinopathy, both in-doamin and country shift. Again we see the trend of performance is similar as compared to Table 2 in the paper with accuracy improvements of aorund 0.3-0.5% for most cases and 0.2-0.3 % decrease for a few cases. However, the performance gap is similar to using the ResNet-50 baseline. Similarly, the trend for Imagenet-100 (Table 19) is approximately same as the paper with errors improved in the range 0.3-0.8 as compared to Table 3 in the paper. This change is visible for all the methods. This can lead to a conclusion that architecture might not be playing a major role in analyzing relative performance for selective classification. However, any concrete claims require a more rigorous testing with various *state-of-the-art* architectures proposed recently.

Table 18: Comparison on the Kaggle dataset and the APTOS dataset under the country shift setup using WRN-28-10 backbone.

| | Kaggle Dataset (*in-distribution*) | | | | | | | | | |
|---|---|---|---|---|---|---|---|---|---|---|
| | 40% retained | | 50% retained | | 60% retained | | 80% retained | | 100% retained | |
| | AUC(%) | Acc.(%) | AUC(%) | Acc.(%) | AUC(%) | Acc.(%) | AUC(%) | Acc.(%) | AUC(%) | Acc.(%) |
| SN | $95.9 \pm 0.1$ | $97.2 \pm 0.1$ | $94.1 \pm 0.1$ | $96.2 \pm 0.2$ | $93.8 \pm 0.2$ | $95.5 \pm 0.2$ | $92.6 \pm 0.1$ | $93.8 \pm 0.1$ | $91.6 \pm 0.3$ | $91.2 \pm 0.2$ |
| SAT | $96.8 \pm 0.1$ | $97.2 \pm 0.1$ | $95.3 \pm 0.2$ | $96.2 \pm 0.1$ | $94.1 \pm 0.3$ | **95.8 ± 0.1** | $93.1 \pm 0.1$ | $93.8 \pm 0.2$ | **92.1 ± 0.3** | **91.6 ± 0.2** |
| Ours | **97.7 ± 0.1** | **98.5 ± 0.0** | **96.7 ± 0.2** | **97.1 ± 0.1** | **94.2 ± 0.3** | $95.2 \pm 0.1$ | **93.2 ± 0.2** | **94.2 ± 0.2** | $91.8 \pm 0.3$ | $91.3 \pm 0.1$ |
| | APTOS Dataset (*country shift*) | | | | | | | | | |
| SN | $86.4 \pm 0.3$ | $88.1 \pm 0.5$ | $87.7 \pm 0.1$ | $88.8 \pm 0.3$ | $90.1 \pm 0.4$ | $87.9 \pm 0.2$ | $91.6 \pm 0.4$ | $87.8 \pm 0.2$ | **93.5 ± 0.1** | $86.9 \pm 0.1$ |
| SAT | $87.5 \pm 0.2$ | $89.6 \pm 0.4$ | $88.2 \pm 0.2$ | $89.4 \pm 0.2$ | $89.4 \pm 0.2$ | $88.2 \pm 0.1$ | $91.6 \pm 0.3$ | $87.3 \pm 0.2$ | $92.9 \pm 0.3$ | **87.3 ± 0.2** |
| Ours | **89.2 ± 0.5** | **91.2 ± 0.2** | **90.5 ± 0.3** | **91.1 ± 0.3** | **91.2 ± 0.2** | **90.4 ± 0.2** | **92.2 ± 0.3** | **88.4 ± 0.3** | $92.5 \pm 0.4$ | $86.4 \pm 0.2$ |

Table 19: Comparison on the ImageNet-100 dataset using the WRN-28-10 backone.

| | 0.6 | 0.7 | 0.8 | 0.9 | 1 |
|---|---|---|---|---|---|
| SN | $2.04 \pm 0.2$ | $3.32 \pm 0.1$ | $5.89 \pm 0.2$ | $8.96 \pm 0.1$ | $13.07 \pm 0.3$ |
| SAT | $1.76 \pm 0.1$ | $2.56 \pm 0.2$ | $5.07 \pm 0.3$ | $8.63 \pm 0.2$ | **13.02 ± 0.4** |
| Ours | **1.42 ± 0.1** | **2.11 ± 0.3** | **4.78 ± 0.4** | **8.17 ± 0.3** | $13.23 \pm 0.3$ |

## F.3 COMPUTATIONAL COMPLEXITY, CONVERGENCE, TRAINING COST:

*Empirical cost.* Per training epoch, we take around 1.2x the naive baseline's running time. The total number of epochs required are 1.2x - 1.5x of ERM classifier. This makes the training process on average 1.5 times more expensive.

These findings were consistent across a wide range of datasets and ranges of hyperparameters, supporting a modest, deterministic increase in running time. This increase is comparable to some selective classification baselines, e.g., 1.3x increase in epochs for SN and 1.2x for SAT. Note, in addition, that baselines such as SAT and DG only work for a pre-determined budget, and changing budget requires retraining from scratch. We only require a one-time training cost.

Table 20: Analyzing the relative time required, w.r.t. ERM, by our method for various datasets (under various setups) used in the paper.

|  | IN-1K | CIFAR-100 | DR | IN-100 | Camelyon | iWildCam |
|---|---|---|---|---|---|---|
| Time per epoch | 1.2 | 1.2 | 1.2 | 1.2 | 1.2 | 1.2 |
| Num epochs | 1.2 | 1.4 | 1.5 | 1.3 | 1.4 | 1.5 |

*Convergence.* The Meta-Weight-Net paper (Appendix C) proves convergence of a general bi-level optimization formulation under specific assumptions – namely that the training and meta-loss are lipschitz smooth with bounded gradient. These conditions apply to our meta-loss as well, and the convergence guarantees also transfer.

### F.4 CONTROLLING FOR U-SCORE PARAMETERS

To control for the extra parameters used by U-SCORE, we compare all baselines trained on ResNet-101 (44M), with our method trained on ResNet-50 (23M) + ResNet-18 (11M) meta-network for our method. The AUARC (Area under accuracy rejection curve) metrics are provided below. With noticeably fewer parameters, we still outperform the baselines.

Table 21: Controlling for extra parameters used by U-SCORE.

| Method | Architecture | DR (OOD) | ImageNet-1K |
|---|---|---|---|
| MCD | RN101(44M) | 88.89 | 87.3 |
| DG | RN101(44M) | 88.87 | 87.6 |
| SN | RN101(44M) | 88.91 | 87.1 |
| SAT | RN101(44M) | 89.08 | 87.3 |
| Ours | RN50+RN18(34M) | 89.94 | 88.2 |

## G ADDITIONAL BASELINES

### G.1 COMPARISON WITH ENTROPY BASED REGULARIZATION

A recent work (Feng et al., 2022) proposed using the maximum logit directly as the uncertainty measure, of the methods trained with selective classification/learning to abstain objectives, instead of their predicted scores. So for a given method, *e.g.* SelectiveNet(Geifman & El-Yaniv, 2019), it just combines classifier trained with that method with Softmax Response (SR)(Geifman & El-Yaniv, 2017) at response time. It further proposes an entropy regularization loss in addition to cross entropy loss to penalize low max-logit scores. We now analyze the effect of this entropy regularization on the selective classification baselines and our method, comparing them for Kaggle Diabetic Retinopathy data (kag, 2015) used in the paper. For the baselines at the inference time, we follow the strategy proposed in this method, using SR, whereas for ours we go with the U-SCORE at the inference time. Table 22 shows the analysis for this experiment. It can be observed that our method (with the U-SCORE ) is still significantly more effective when trained with entropy regularization as compared to these baselines. Also, using variance reduction based prediction scores are a better criteria as compared to directly applying SR technique for these selective classifiers.

### G.2 COMPARISON WITH RE-WEIGHTING METHODS

As explained in the paper, these methods are train-time-only reweightings, since they learn free parameters for each training instance (Ren et al., 2018; Zhang & Pfister, 2021), or as a function of instance loss (requiring true label) (Shu et al., 2019). In contrast, we learn a neural network which can readily be applied on unseen instances As a compromise, we used (Ren et al., 2018; Shu et al., 2019; Zhang & Pfister, 2021) for training the classifier, and used MCD at test-time for selective classification; this tells us if the training procedure in these results in better classifiers. For ours,

Table 22: Comparison on the Kaggle dataset and the APTOS dataset under the country shift setup trained using entropy regularizer and then selecting based on SR.

| | Kaggle Dataset (*in-distribution*) | | | | | | | | | |
|---|---|---|---|---|---|---|---|---|---|---|
| | 40% retained | | 50% retained | | 60% retained | | 80% retained | | 100% retained | |
| | AUC(%) | Acc.(%) | AUC(%) | Acc.(%) | AUC(%) | Acc.(%) | AUC(%) | Acc.(%) | AUC(%) | Acc.(%) |
| SN | $96.1 \pm 0.1$ | $97.2 \pm 0.1$ | $94.8 \pm 0.2$ | $96.6 \pm 0.1$ | $93.9 \pm 0.2$ | $95.5 \pm 0.1$ | $93.1 \pm 0.2$ | $93.8 \pm 0.1$ | $92.4 \pm 0.3$ | $91.7 \pm 0.3$ |
| SAT | $96.8 \pm 0.1$ | $98.0 \pm 0.1$ | $95.7 \pm 0.1$ | $97.1 \pm 0.1$ | $94.2 \pm 0.2$ | $95.7 \pm 0.1$ | $\mathbf{93.5 \pm 0.3}$ | $\mathbf{94.1 \pm 0.2}$ | $\mathbf{92.6 \pm 0.3}$ | $\mathbf{91.8 \pm 0.2}$ |
| Ours | $\mathbf{97.7 \pm 0.1}$ | $\mathbf{98.6 \pm 0.0}$ | $\mathbf{96.9 \pm 0.2}$ | $\mathbf{97.8 \pm 0.1}$ | $\mathbf{94.8 \pm 0.2}$ | $\mathbf{95.9 \pm 0.3}$ | $92.9 \pm 0.2$ | $93.8 \pm 0.2$ | $92.1 \pm 0.2$ | $91.5 \pm 0.2$ |
| | APTOS Dataset (*country shift*) | | | | | | | | | |
| SN | $87.4 \pm 0.4$ | $89.7 \pm 0.4$ | $89.8 \pm 0.3$ | $89.5 \pm 0.2$ | $89.7 \pm 0.3$ | $87.5 \pm 0.1$ | $91.1 \pm 0.2$ | $87.2 \pm 0.1$ | $\mathbf{93.2 \pm 0.2}$ | $86.44 \pm 0.1$ |
| SAT | $88.3 \pm 0.4$ | $90.9 \pm 0.3$ | $90.2 \pm 0.2$ | $90.3 \pm 0.2$ | $89.8 \pm 0.2$ | $88.5 \pm 0.2$ | $91.8 \pm 0.3$ | $87.7 \pm 0.2$ | $92.6 \pm 0.3$ | $\mathbf{86.7 \pm 0.2}$ |
| Ours | $\mathbf{90.1 \pm 0.5}$ | $\mathbf{92.5 \pm 0.3}$ | $\mathbf{91.4 \pm 0.3}$ | $\mathbf{91.9 \pm 0.3}$ | $\mathbf{91.7 \pm 0.2}$ | $\mathbf{89.9 \pm 0.1}$ | $\mathbf{92.4 \pm 0.3}$ | $\mathbf{88.9 \pm 0.2}$ | $92.4 \pm 0.3$ | $86.2 \pm 0.2$ |

we still use our meta-network to select the instances to classify. The Area under accuracy rejection curve (AUARC metric) is provided in table 23 (under No Var Min in Baselines). It can be observed that our method significantly outperform these methods. To further differentiate the contributions of our U-SCORE, and our meta-loss, we add our variance minimization loss to these re-weighting schemes and also report the results in table 23 (under Var Min). Still our method performs the best thereby proving that both our contributions, *i.e.*, *instance-conditioning* and *variance minimization* hold significant importance in performance improvement.

Table 23: Comparison of our methods with other re-weighting methods based on bi-level optimization on selective classification, calibration in the in-domain val set scenario, with and without adding our proposed variance minimization to their val set objectives.

| | No Var Min in Baselines | | Var Min | | | |
|---|---|---|---|---|---|---|
| | AUARC | | AUARC | | ECE | |
| Method | DR (OOD) | ImageNet-1K | DR (OOD) | ImageNet-1K | DR (OOD) | ImageNet-1K |
| MCD | 88.27 | 87.30 | 88.27 | 87.30 | 9.1 | 3.0 |
| MWN | 88.19 | 87.20 | 88.38 | 87.40 | 8.2 | 3.0 |
| L2R | 88.07 | 87.20 | 88.33 | 87.40 | 8.4 | 3.1 |
| FSR | 88.38 | 87.30 | 88.67 | 87.50 | 8.2 | 2.9 |
| Ours | 89.94 | 88.20 | 89.94 | 88.20 | 6.4 | 2.6 |

### G.3 COMPARISON WITH SIMPLE CALIBRATORS

We compared against ProbOut, Platt scaling, and also its single single parameter version (temperature scaling) which was shown to be better at calibration (Guo et al., 2017). We report the mean and std (AUARC) of 5 different runs. Results on all the datasets are provided in table 24. Our method is able to provide significant gains (upto 2.3%) as compared to all of these methods.

Table 24: Comparison of our method with widely popular and simple calibration schemes.

| Dataset | Probout | Platt scaling | Temp Scaling | Confomal prediction | Ours |
|---|---|---|---|---|---|
| ImageNet-1K | $86.9 \pm 0.1$ | $86.8 \pm 0.1$ | $87.1 \pm 0.1$ | $86.6 \pm 0.2$ | $88.2 \pm 0.2$ |
| ImageNet-100 | $92.30 \pm 0.1$ | $92.07 \pm 0.2$ | $92.25 \pm 0.2$ | $91.70 \pm 0.1$ | $94.50 \pm 0.2$ |
| CIFAR-100 | $91.20 \pm 0.2$ | $91.10 \pm 0.1$ | $91.35 \pm 0.1$ | $90.95 \pm 0.2$ | $93.20 \pm 0.1$ |
| DR (OOD) | $87.08 \pm 0.2$ | $86.75 \pm 0.2$ | $86.96 \pm 0.2$ | $86.90 \pm 0.2$ | $89.40 \pm 0.1$ |
| DR (ID) | $92.25 \pm 0.1$ | $91.90 \pm 0.1$ | $92.55 \pm 0.1$ | $91.55 \pm 0.2$ | $94.12 \pm 0.1$ |

### G.4 CORRELATION BETWEEN INSTANCE WEIGHT AND PREDICTIVE ENTROPY

We calculated the correlation between weights and predictive entropy in table. Further, we also evaluated entropy itself as an uncertainty measure. The results are provided in table 25. The correlations are substantial, conforming to the claim that we capture model uncertainty. However,

we outperform entropy, suggesting that entropy is by itself not the gold standard for uncertainty measurement, and a 100% correlation with it is not desirable.

Table 25: Comparing entropy as an uncertainty measure against our U-SCOREand also calculating the correlation between the two.

|  | DR-In-D | DR(OOD) | CIFAR-100 | Im-100 | Im-1k |
|---|---|---|---|---|---|
| Entropy based | 92.91 | 87.93 | 92.25 | 93.15 | 87.05 |
| Ours | 94.12 | 89.94 | 93.20 | 94.50 | 88.20 |
| Correlation | 0.57 | 0.61 | 0.59 | 0.63 | 0.68 |

## H  CONTROLS FOR SELECTIVE CLASSIFICATION

### H.1  SELECTIVE CLASSIFICATION ON HARD SAMPLES

A concern with selective classification might be that significant initial gains may be obtained by quickly rejecting only the (rare) hard sasmples, while ranking the remaining examples poorly. To control for this, we compared selective accuracy (Area under accuracy rejection curve) for Imagenet-trained classifiers on the naturally occurring hard-example test set Imagenet-A. In this test set, all samples are in some sense hard samples, and there are no shortcuts to good selective classification accuracy. The results are provided in table 26 Even among hard samples, our method is able to better order instances according to uncertainty.

Table 26: Comparison of our methods and the baselines for selective classification on the ImageNet-A dataset at various coverages.

| Method | MCD | DG | SN | SAT | Ours |
|---|---|---|---|---|---|
| ImageNet-A | 8.44 | 8.53 | 8.64 | 8.91 | 9.98 |

### H.2  MATCHED TEST SETS FOR SELECTIVE CLASSIFICATION

Another challenge in selective classification is that each method can choose to reject different instances, and end up reporting accuracy on nonidentical sets of data. To control for this, we use the ImageNet-A dataset for testing so that the test set comprises mostly hard examples. We apply our selection method using the U-SCORE to select examples for each coverage and then test our classifier as well as the other baselines' classifier on the same set of chosen examples. The results are reported in table 27. The column PSS (previous Selection scheme) denotes the result of previous comparison whereas column OSS (our selection scheme) denotes the result when our selection scheme for each of the baseline training methods is used. The results show that our selection scheme is capable of identifying less erroneous examples quite better than other selection schemes, since our selection improves each method's accuracy. Further, our classifier is also more accurate on the selected set, suggesting two separate sets of benefits from our method. Our U-SCORE can identify erroneous examples (intrinsically hard examples) better than other methods – this is a measure of uncertainty. The modeled uncertainty is of course best for the classifier jointly trained with it but is partially applicable to other classifiers too.

The AUARC metric is as follows:

## I  SHARED WEIGHTS

We now analyze the effect of sharing parameters between the meta-network and the classifier. Here the classifier encoder is used by the meta-network upto the fully connected layer and then a separate K-layered fully connected neural network is used as the U-SCORE. We call this version as Ours (shared). The results for selective classification task with different values of K are provided in in table 28 (comprising the AUARC metric).

Table 27: Analyzing the scenario when our selection scheme (OSS) is applied to select examples in the ImageNet-A dataset for selective classification and then baseline trained methods along with ours, all are evaluated on this selected set of examples. This is also compared against the case when the selection for baseline is done using their respective selection scheme (PSS).

|  | MCD | DG | SN | SAT | Ours |
|---|---|---|---|---|---|
| PSS | 8.44 | 8.53 | 8.64 | 8.91 | 9.98 |
| OSS | 8.87 | 8.95 | 8.46 | 9.12 | 9.98 |

|  | DR (ID) | DR (OOD) | CIFAR-100 | ImageNet-100 | ImageNet-1K |
|---|---|---|---|---|---|
| Ours | $94.12 \pm 0.1$ | $89.94 \pm 0.1$ | $93.20 \pm 0.1$ | $94.95 \pm 0.1$ | $88.20 \pm 0.2$ |
| Ours (Shared, K=4) | $93.05 \pm 0.1$ | $88.07 \pm 0.1$ | $92.03 \pm 0.1$ | $93.67 \pm 0.1$ | $86.85 \pm 0.1$ |
| Ours (Shared, K=6) | $93.12 \pm 0.1$ | $88.13 \pm 0.1$ | $92.16 \pm 0.1$ | $93.32 \pm 0.1$ | $86.75 \pm 0.1$ |
| Ours (Shared, K=8) | $92.98 \pm 0.1$ | $88.02 \pm 0.1$ | $91.98 \pm 0.1$ | $93.10 \pm 0.1$ | $86.68 \pm 0.1$ |

Table 28: Comparison with variant having shared weights for U-SCORE and classifier

It can be observed that this shared weights setup is somewhat sub-optimal in nature. Here the encoder part is only updated w.r.t. predictor loss, otherwise the optimization becomes unstable given the bi-level objective. Only the K layers are updated using the loss for the U-SCORE. A potential explanation is that U-SCORE uses a markedly different set of features from the image compared to the primary classifer–the classifier needs to find discriminative features, whereas the U-SCORE needs to identify features, potentially across classes, that flag instance hardness. We also tried other configurations where we tried attaching the meta-network Fully-connected neural network to different intermediate representations of the classifier encoder architecture but no improvements were observed.

## J    FINE-TUNING ON VAL SET FOR OOD SETUPS

We now analyze another baseline against our method in the OOD setup. Here instead of our bi-level optimization setup, ERM trained baseline is directly tuned on the validation set. We have three kinds of OOD setups in the paper: one where the train, val and test are from different domains (iWildCam, Camelyon). Second where the train and val from same domain and test from different DR (OOD). Third, where we have some unlabelled examples available from the same domain as the OOD test set and train, val sets are in-domain. We now analyze the above discussed baseline on setups 1 and 3 where we can have OOD examples in the validation set. In none of the setups, we have labelled examples from the test domain. Thus, we can't naively tune ERM on the test domain. For the first setup, we directly fine-tune the ERM trained classifier on the validation set alongside unlabelled example from the test set using our unsupervised variance minimization objective. For the third setup, since the OOD examples are unlabelled, we instead tune the ERM trained classifier using our unsupervised variance minimization objective. The results are provided in table 29. For selective classification we use MCD on top this baseline. We term this baseline as MCD (val-tuned).

|  | Camelyon | iWildCam | DR (OOD) |
|---|---|---|---|
| REVAR | 76.32 | 77.98 | 89.94 |
| REVAR-PV | 78.12 | 79.86 | 91.23 |
| MCD (val-tuned) | 75.25 | 76.34 | 88.06 |

Table 29: Comparison with a validation set tuned baseline.

It can be observed that the MCD(val tuned) baseline performs worse than ours in all cases, further advocating the usefulness of our proposed scheme.

## K  ERROR BARS FOR TABLES

In the main paper, due to lack of space, some tables (4-8) donot contain the std values from the five runs of each exp. Only mean values are reported. We provide the std values alongside the mean values in the corresponding tables below.

Table 30: **AUARC:** In-Domain, OOD test set results with error bars

| | ImageNet-A | | | ImageNet-C | | | ImageNet-R | |
| Ours | MCD | SAT | Ours | MCD | SAT | Ours | MCD | SAT |
|---|---|---|---|---|---|---|---|---|
| **9.98**± 0.05 | 8.44 ± 0.07 | 8.91 ± 0.08 | **65.9** ± 0.1 | 63.7 ± 0.2 | 64.2 ± 0.1 | **68.8** ± 0.1 | 66.8 ± 0.1 | 67.1 ± 0.1 |

Table 31: **AUARC:** OOD val, test set results with error bars   Table 32: **Label Noise**: Plex Model

| Data | MCD | SAT | Revar | Revar-PV |
|---|---|---|---|---|
| Camelyon | 74.99 ± 0.1 | 75.16 ± 0.1 | 76.32 ± 0.2 | 78.12 ± 0.1 |
| iWildCam | 76.07 ± 0.1 | 76.17 ± 0.0 | 77.98 ± 0.1 | 79.86 ± 0.2 |

| | Plex | Plex+ours |
|---|---|---|
| IN-100H | 0.75 ± 0.001 | **0.71** ± 0.002 |
| CF-100H | 0.49 ± 0.001 | **0.47**± 0.001 |

Table 33: **Label Noise**: Re-weighting methods result with error bars

| | MCD | MWN | L2R | FSR | Ours |
|---|---|---|---|---|---|
| Inst.CIFAR-100 | 61.12 ± 0.1 | 65.89 ± 0.1 | 67.12 ± 0.2 | 70.21 ± 0.1 | **71.87**± 0.1 |
| Clothing1M | 68.78± 0.1 | 73.56± 0.0 | 72.97± 0.1 | 73.86± 0.1 | **73.97**± 0.2 |

## L  FURTHER ANALYSIS FOR SELECTIVE CLASSIFICATION WITH OTHER ARCHITECTURES

We further analyze our method with the ViT-Small, Wide ResNet-28-10 (WRN-28-10) architectures for the classifier by replicating Tab. 2, comprising the area under accuracy rejection curve (AUARC) metric, from the main paper using them. Tab. 34, Tab. 35 denote the tables corresponding to these architectures respectively. The U-SCORE is same as the one used for Tab. 2. It can be observed that, upon going to these bigger architectures, there is either a minor improvement in performance gains or they remain same as Tab. 2. This further verifies the effectiveness of our method and advocates for its usefulness for various tasks requiring different kinds of neural network architectures.

Table 34: Selective classification with ViT-Small architecture: AUARC metric.

| | Selective Classification Baselines | | | | | New Baselines | | REVAR |
| | SR | MCD | DG | SN | SAT | VR | MBR | Ours |
|---|---|---|---|---|---|---|---|---|
| DR(In-Dist.) | 93.98 ± 0.1 | 94.86 ± 0.0 | 94.41 ± 0.1 | 94.56 ± 0.2 | 94.61 ± 0.1 | 93.98 ± 0.1 | 94.12 ± 0.1 | **95.55 ± 0.1** |
| DR(OOD) | 88.52 ± 0.2 | 89.31 ± 0.1 | 88.91 ± 0.2 | 89.32 ± 0.2 | 89.86 ± 0.1 | 88.89 ± 0.2 | 88.94 ± 0.2 | **90.86 ± 0.1** |
| CIFAR-100 | 93.22 ± 0.1 | 93.44 ± 0.1 | 93.02 ± 0.1 | 88.79 ± 0.1 | 93.46 ± 0.2 | 92.86 ± 0.2 | 92.99 ± 0.2 | **94.19 ± 0.1** |
| ImageNet-100 | 94.25 ± 0.1 | 95.12 ± 0.1 | 94.62 ± 0.2 | 94.40 ± 0.2 | 94.82 ± 0.1 | 94.15 ± 0.2 | 94.76 ± 0.2 | **95.98 ± 0.1** |
| ImageNet-1K | 87.50 ± 0.2 | 88.70 ± 0.1 | 88.10 ± 0.1 | 88.40 ± 0.2 | 88.20 ± 0.2 | 88.25 ± 0.2 | 87.65 ± 0.2 | **89.50 ± 0.1** |

Table 35: Selective classification with WRN-28-10 architecture: AUARC metric.

| | Selective Classification Baselines | | | | | New Baselines | | REVAR |
| | SR | MCD | DG | SN | SAT | VR | MBR | Ours |
|---|---|---|---|---|---|---|---|---|
| DR(In-Dist.) | 92.95 ± 0.2 | 93.67 ± 0.1 | 93.22 ± 0.2 | 93.42 ± 0.2 | 93.65 ± 0.1 | 92.67 ± 0.1 | 93.02 ± 0.1 | **94.51 ± 0.2** |
| DR(OOD) | 87.75 ± 0.2 | 88.40 ± 0.1 | 88.26 ± 0.1 | 88.68 ± 0.2 | 89.12 ± 0.2 | 88.08 ± 0.1 | 88.22 ± 0.2 | **90.17 ± 0.1** |
| CIFAR-100 | 92.71 ± 0.2 | 92.95 ± 0.2 | 92.58 ± 0.2 | 88.67 ± 0.2 | 93.05 ± 0.1 | 92.49 ± 0.1 | 92.80 ± 0.1 | **93.68 ± 0.1** |
| ImageNet-100 | 93.25 ± 0.1 | 94.48 ± 0.1 | 93.76 ± 0.1 | 93.88 ± 0.2 | 94.41 ± 0.1 | 93.51 ± 0.1 | 94.06 ± 0.1 | **95.38 ± 0.1** |
| ImageNet-1K | 86.46 ± 0.1 | 87.67 ± 0.0 | 87.35 ± 0.2 | 87.28 ± 0.2 | 87.45 ± 0.1 | 87.28 ± 0.1 | 86.60 ± 0.1 | **88.63 ± 0.2** |

