training set; this follows a long line of work which has used bilevel optimization to address covariate shift from train to validation through weighted loss formulations at instance level (Sugiyama et al., 2008) and group level (Mohri et al., 2019).

### 3.3 INSTANCE-CONDITIONAL WEIGHTS IN REVAR

Our goal in REVAR is to learn an instance-conditional scorer U-SCORE that is used both for train-time reweighting, and at test time as a measure of predictive uncertainty. Previous work on bilevel optimization for reweighting (Shu et al., 2019; Ren et al., 2018; Zhang & Pfister, 2021) cannot be used at test time because the learned weights are free parameters (Ren et al., 2018) or a function of instance loss (Shu et al., 2019; Zhang & Pfister, 2021). We address this challenge by learning instance weights as a direct function of the instance itself, i.e., $w = g_\Theta(x)$, allowing us to capture a much richer and unconstrained measure of model uncertainty, which is also robustly estimated using the bilevel formulation. Our bilevel formulation now becomes:

$$\theta^* = \arg\min_\theta \frac{1}{N} \sum_{i=1}^N g_\Theta(x_i) \cdot l(y_i, f_\theta(x_i)) \quad s.t. \ \Theta^* = \arg\min_\Theta \mathcal{L}_{meta}(X^s, Y^s, \theta^*) \quad (3)$$

### 3.4 VARIANCE MINIMIZATION AS META-REGULARIZATION

We now define the meta-loss $\mathcal{L}_{meta}$ and in particular, a novel variance-minimizing regularizer that substantially improves the ability of $g(\cdot)$ to capture model uncertainty. This meta-regularizer $l_{eps}(\theta, x)$ is added to the cross-entropy classification loss $l(y, f_\theta(x))$ that is typically part of the meta-loss, leading to the following meta-objective on the specialized set $(\mathcal{X}^s, \mathcal{Y}^s)$:

$$\mathcal{L}_{meta} = \mathcal{L}_c(X_s, Y_s) + \mathcal{L}_{eps}(\theta, X_s) = \sum_{j=1}^M l(y_j^s, f_\theta(x_j^s)) + l_{eps}(\theta, x_j^s) \quad (4)$$

**Minimizing Bayesian Posterior uncertainty:** We take inspiration from the Bayesian NN literature which regularizes the posterior on weight distribution so as to avoid overfitting or to embed extra domain knowledge. Unlike standard neural networks which output a point estimate for a given input, Bayesian networks (Buntine, 1991; Tishby et al., 1989; Denker et al., 1987; Blundell et al., 2015; Kwon et al., 2020) learn a distribution $p(\omega|D)$ over the neural network weights $\omega$ given the dataset $D$ using maximum a posteriori probability (MAP) estimation. The predictive distribution for the output $y^*$, given the input $x$ and $D$, can be then calculated by marginalisation as follows: $p(y^*|x^*, D) = \int p(y^*|x^*, \omega)p(\omega|D)d\omega \approx \frac{1}{K} \sum_{k=1}^K p(y^*|x^*, \omega^k)$. Here we utilize a recent result (Gal & Ghahramani, 2016) that augmenting the training of a deterministic neural network with dropout regularization yields a variational approximation for a Bayesian Neural Network. At test time, taking multiple forward passes through the neural network for different dropout masks yields a *Monte-Carlo* approximation to Bayesian inference, and thereby a predictive distribution. The variance over these Monte Carlo samples is therefore a measure of predictive uncertainty:

$$l_{eps}(\theta, x) \approx \frac{1}{K} \left( \sum_{k=1}^K (f_{\mathcal{D}_k \odot \theta}(x) - E[f_{\mathcal{D}_k \odot \theta}(x)])^2 \right) \quad (5)$$

where $\mathcal{D}_k$ denotes the dropout mask at $k^{th}$ sample and $\mathcal{D}_k \odot \theta$ denotes the application of this dropout mask to the neural network parameters. This MCD measure is popular as an estimate of instance uncertainty (Gal & Ghahramani, 2016), and is competitive with state-of-the-art methods for selective classification (Filos et al., 2019).

We propose to use this variance-based estimate of posterior uncertainty as a *meta-regularization term* in our approach. In particular, this means that instead of directly minimizing the posterior uncertainty on the training data w.r.t. primary model parameters $\theta$, we minimize it w.r.t. U-SCORE parameters $\Theta$ on the specialized set instead. This approach provides a significant incentive for the U-SCORE to accurately capture the various notions of uncertainty in the data.

### 3.5 META-LEARNING WITH BILEVEL LOSS

The modeling choices we have laid out above result in a bi-level optimization scheme involving the meta-network and classifier parameters. This is because the values of each parameter set $\theta$ and $\Theta$

influence the optimization objective of the other. Expressing $\mathcal{L}_{meta}$ as a function $\mathcal{L}$ of inputs $X^s$, $Y^s$, $\theta$, this bi-level optimization scheme can be formalized as:

**Calculating updates**. Instead of solving completely for the inner loop (optimizing $\Theta$) for every setting of the outer parameter $\theta$, we aim to solve this bilevel optimization using alternating stochastic gradient descent updates. At a high level, the updates are:

$$\Theta_{t+1} = \Theta_t - \alpha_1 \nabla_\Theta \mathcal{L}(X^s, Y^s, \theta_t) \quad ; \quad \theta_{t+1} = \theta_t - \alpha_2 \frac{1}{N} \nabla_\theta \left( \sum_{i=1}^N g_\Theta(x_i) \cdot l(y_i, f_\theta(x_i)) \right) \quad (6)$$

where $\alpha_1$ and $\alpha_2$ are the learning rates corresponding to these networks, $\mathcal{L}_c$ is the classification loss on the dataset $(X,Y)$ using a network with params $\theta$, and $G_\Theta$ is the vector of weights predicted by $g_\Theta$ for each input sample in $X$, with $\Theta := \Theta_{t+1}$. This style of stochastic optimization is commonly used for solving bilevel optimization problems in a variety of settings (Algan & Ulusoy, 2020; Shu et al., 2019). Further details, including all approximations used for deriving these equations, are provided in the appendix.

---

**Algorithm 1** REVAR training procedure.

---

**Require:** Prediction Network parameters $\theta$, U-SCORE parameters $\Theta$, learning rates $(\beta_1, \beta_2)$, dropout rate $p_{drop}$, training data $\{x_i, y_i\}_{i=1}^N$, validation data $\{x_i^s, y_i^s\}_{i=1}^M$, U-SCORE update interval $\mathcal{M}$ .
**Ensure:** Robustly trained classifier parameters $\theta^*$, U-SCORE parameters $\Theta^*$ to predict uncertainty.
1: Randomly initialize $\theta$ and $\Theta$, $t = 1$;
2: **for** e = 1 **to** E **do**                                                   $\triangleright$ E: number of epochs
3:     sample a minibatch $\{(x_i, y_i)\}_{i=1}^n$ from training data;         $\triangleright$ n denotes the batch size
4:     **if** $t \% \mathcal{M} == 0$ **then**
5:         Create a copy of the current prediction model, denoting parameters by $\hat{\theta}$
6:         sample minibatch $\{(x_i^v, y_i^v)\}_{i=1}^m$ from validation data
7:         $\hat{\theta} \leftarrow \hat{\theta} - \beta_1 \nabla_{\hat{\theta}} \sum \left( l(f_{\hat{\theta}}(x), y) \right)$         $\triangleright$ Update the copy of prediction model
8:         $\Theta \leftarrow \Theta - \beta_2 \nabla_\Theta \sum \left( l(y_i^v, f_{\hat{\theta}}(x_i^v)) + l_{eps}(\hat{\theta}, x_i^v) \right)$     $\triangleright$ Update U-SCORE using Eq. 5
9:     **end if**
10:    $\theta \leftarrow \theta - \beta_1 \nabla_\theta \sum g_\Theta(x_i) l(f_\theta(x_i), y_i)$;         $\triangleright$ Update the prediction model
11:    $\theta^* \leftarrow \theta$; $\Theta^* \leftarrow \Theta$; $t \leftarrow t + 1$
12: **end for**

---

## 4 U-SCORE CAPTURES DIFFERENT SOURCES OF UNCERTAINTY

We now create a set of synthetic generative models for linear regression and study the performance of our algorithm for conceptual insights. We investigate three kinds of uncertainty that depends on the input instance $x$: 1) Samples that are atypical with respect to train but typical with respect to validation 2) Samples where label noise is higher 3) Samples where uncertainty in the label is due to some unobserved latent features that affect the label.

Usually (1) and (3) are considered to be "epistemic" uncertainty and (2) would fall under "aleatoric" uncertainty. (1) is due to covariate shift and (3) is due to missing features relevant for the label. Surprisingly, we show in this section is that *our algorithm's weights are proportional to uncertainty from (1) while being inversely proportional to uncertainty of type (2) and (3).* This is also desirable from a theoretical perspective, as we explain below–for instance, when (1) and (3) are absent, the best solution is to downweight samples with larger label noise (Das et al., 2023). Similarly, one would desire examples that are typical with respect to validation and atypical with respect to train to be weighted higher when only (1) is present. We show that our algorithm captures these notions, and furthermore smoothly interpolates between them depending on the mix of different sources of uncertainty.

**Generative Model:** For all the results in this section, for both training and validation data for all , $Y$ is sampled as follows.

$$Y = W_{\text{data}}^T X + (\mathcal{N}(0, 1) \cdot [c + G^T X]) \quad (7)$$

$X \in \mathbb{R}^{72 \times 1}$. $X = [X_c X_e], X_c \in \mathbb{R}^{48 \times 1}, X_e \in \mathbb{R}^{24 \times 1}$. For training data, we sample $X^{\text{train}} \sim \mathcal{N}(\mu, \Sigma)$. For validation, $X^{\text{val}} \sim \mathcal{N}(\mu', \Sigma)$ where $\mu' = \mu + s\mathcal{N}(\mu_s, \Sigma_s)$; here $s > 0$ is a scalar that

determines the amount of covariate shift between training and validation. $W_{\text{data}}^T = [W_c^T \; W_e^T]$ where $W_c \in \mathbb{R}^{48 \times 1}$, $W_e \in \mathbb{R}^{24 \times 1}$.

**Evaluation, Baselines & Metrics:** We train our method on paired train-validation datasets sampled according to different scenarios, and inspect U-SCORE scores for points $x$ in the training set. For each scenario, we predict a theoretical ideal for the instance dependent weights, and calculate $R^2$ score of model fits for the U-SCORE outputs against the theoretical ideal. We compare against MWN Shu et al. (2019), a baseline that calculates loss-dependent instance weights using bilevel optimization. We also measure the specific contributions of our variance-minimization regularization, by evaluating a second baseline that is identical to REVAR except for this meta-regularization term–we term this Instance-Based Reweighting (IBR).

Table 1: $R^2$ metric. $\lambda_1, \lambda_2$ are fitting coefficients. $h$ (hardness) is Euclidean distance from training data mean $h = (x - \mu)^2$, and captures magnitude of covariate shift. Other terms quantify sample dependent label noise.

| S | Target | MWN | IBR | Ours |
|---|--------|-----|-----|------|
| 1 | $\frac{\lambda_1}{|G^T X|^2}$ | 0.77 | 0.78 | 0.84 |
| 2 | $\frac{\lambda_1}{|G^T X|^2} + \lambda_2 \cdot h$ | 0.58 | 0.62 | 0.80 |
| 3 | $\frac{\lambda_1}{W_e^T \Sigma(X_e|X_c) W_e}$ | 0.46 | 0.52 | 0.81 |
| 4 | $\frac{\lambda_1}{W_e^T \Sigma(X_e|X_c) W_e} + \lambda_2 \cdot h$ | 0.51 | 0.57 | 0.82 |
| 5 | $\lambda_1 \cdot \mathcal{U}(0, 1)$ | 0.44 | 0.58 | 0.84 |

Figure 1: Scenario 2 and 4 analysis with increasing distribution shift

**Scenario 1 - Sample Dependent Label Noise and No Shift:.** $c = 0, s = 0, G \neq 0$. This represents a scenario where there is no covariate shift but label uncertainty in both train and validation depend on the sample. Label noise scales as $|G^T X|^2$, while weights of the meta-network are *inversely proportional* to this quantity, in a manner supported by theory **?** (Tab. 1).

**Scenario 2 - Sample Dependent Label noise and Covariate Shift:** We set $c = 0, G \neq 0, s \neq 0$. Here the meta network weights roughly follow the relationship given by: $w(x) \sim \frac{\lambda_1}{|G^T x|^2} + \lambda_2 \cdot (x - \mu)^2$. In other words, U-SCORE weights are *inversely proportional to label noise and directly proportional to the uncertainty due to covariate shift.*. Further, weights shift smoothly towards uncertainties from covariate shift as the magnitude of covariate shift increases Sec. 4.

**Scenario 3 - Hardness due to missing relevant features:** We set $c = 1, G = 0, s = 0$. However, only $X_c$ is available to the learner in both train and validation. Therefore, there is no explicit shift, however the missing features $X_e$ influences the label. Interestingly this behaves much like sample-dependent label noise–given the features seen ($X_c$), there is added label noise that can't be fit, proportional to $W_e^T \Sigma(X_e|X_c) W_e$. Indeed, the weights predicted by U-SCORE roughly scales as $\frac{1}{W_e^T \Sigma(X_e|X_c) W_e}$ (Tab. 1). Although conventionally treated as "epistemic uncertainty", our meta network's weights are inversely proportional to this, as desired.

**Scenario 4 - Dropping Features and covariate shift in validation set:.** We set $c = 1, G = 0, s > 0$ and only $X_c$ is available to the learner. In this case, the weights predicted by our meta-network for this setup roughly follows the relationship $\frac{\lambda_1}{W_e^T \Sigma(X_e|X_c) W_e} + \lambda_2 (x - \mu)^2$. Here, U-SCORE *treats uncertainty due to missing features as label noise and its weights are proportional to uncertainty due to shift*. As before, from Sec. 4, weights reflect uncertainties from covariate shift more than label noise as magnitude of shift increases.

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

## 6 CONCLUSION

We proposed a unified approach to modeling uncertainty (REVAR) that reweights training instances for robust learning, and provides superior test-time measures of uncertainty. We proposed a novel variance-minimizing regularization for the meta-objective that is key to effectively capturing a range of precisely defined notions of uncertainty, and for SOTA performance in selective classification, calibration, prediction accuracy across a wide range of datasets, domain-shift challenges, and model architectures including large pre-trained models (PLEX). We are interested in developing a theoretical framework to better understand the relationship between classifier robustness, measures of uncertainty, and variance minimization.

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

|---|---|---|---|---|---|---|---|---|---|---|
| | Kaggle Dataset (*in-distribution*) | | | | | | | | | |
| MCD | 96.3 ± 0.1 | 97.8 ± 0.0 | 95.2 ± 0.1 | 97.1 ± 0.1 | 93.7 ± 0.3 | 95.3 ± 0.2 | 92.3 ± 0.2 | 92.8 ± 0.1 | 91.2 ± 0.2 | 90.6 ± 0.1 |
| DG | 95.9 ± 0.1 | 97.2 ± 0.1 | 94.4 ± 0.2 | 96.4 ± 0.1 | 93.3 ± 0.2 | 95.1 ± 0.1 | 92.5 ± 0.3 | 93.1 ± 0.1 | 91.3 ± 0.3 | 90.8 ± 0.1 |
| SN | 95.8 ± 0.1 | 97.0 ± 0.1 | 94.2 ± 0.2 | 96.1 ± 0.1 | 93.5 ± 0.3 | 95.2 ± 0.1 | 92.8 ± 0.1 | 93.4 ± 0.1 | 91.4 ± 0.3 | 90.9 ± 0.2 |
| SAT | 96.5 ± 0.0 | 97.9 ± 0.0 | 95.0 ± 0.1 | 96.8 ± 0.1 | 93.9 ± 0.2 | **95.6** ± 0.1 | 92.7 ± 0.2 | 93.6 ± 0.2 | **91.7 ± 0.3** | **91.1 ± 0.2** |
| Ours | **97.5 ± 0.1** | **98.4 ± 0.0** | **96.3 ± 0.2** | **97.4 ± 0.1** | **94.4 ± 0.3** | 95.5 ± 0.2 | **92.9 ± 0.2** | **93.8 ± 0.2** | 91.5 ± 0.3 | 91.0 ± 0.1 |
| | APTOS Dataset (*country shift*) | | | | | | | | | |
| MCD | 79.8 ± 0.8 | 87.9 ± 0.5 | 87.2 ± 0.4 | 87.8 ± 0.2 | 89.1 ± 0.2 | 87.3 ± 0.2 | 91.4 ± 0.3 | 86.9 ± 0.2 | 93.6 ± 0.3 | 86.2 ± 0.2 |
| DG | 83.7 ± 0.6 | 87.5 ± 0.3 | 88.1 ± 0.3 | 87.1 ± 0.2 | 90.1 ± 0.6 | 86.9 ± 0.2 | 91.9 ± 0.2 | 86.2 ± 0.2 | **93.7 ± 0.6** | 86.1 ± 0.1 |
| SN | 86.2 ± 0.4 | 88.4 ± 0.4 | 88.1 ± 0.2 | 88.3 ± 0.2 | 89.7 ± 0.3 | 87.5 ± 0.1 | 91.1 ± 0.2 | 87.2 ± 0.1 | 93.2 ± 0.2 | 86.3 ± 0.1 |
| SAT | 87.3 ± 0.3 | 89.8 ± 0.3 | 88.7 ± 0.2 | 89.2 ± 0.2 | 89.3 ± 0.2 | 87.9 ± 0.1 | 91.3 ± 0.3 | 87.1 ± 0.2 | 92.7 ± 0.3 | **86.9 ± 0.2** |
| Ours | **89.2 ± 0.4** | **91.4 ± 0.2** | **90.2 ± 0.3** | **90.7 ± 0.3** | **90.9 ± 0.2** | **89.9 ± 0.1** | **91.8 ± 0.2** | **88.1 ± 0.2** | 92.3 ± 0.3 | 86.1 ± 0.2 |