# OpenReview forum: "Learning model uncertainty as variance-minimizing instance weights"
_ICLR.cc/2024/Conference — ICLR 2024 poster_

### Official Review · Reviewer_1jLE · 2023-10-25

**Soundness:** 3 good
**Presentation:** 3 good
**Contribution:** 3 good
**Rating:** 6
**Confidence:** 3

**Summary:**

The authors propose ReVar technique for predicting model uncertainty at train and test time. In particular, ReVar is notable for aiming to capture various different sources of uncertainty, including covariance shift and high label noise. ReVar uses bi-level optimization to learn both a primary model f(x) and auxiliary uncertainty model g(x), which serves as an instance-dependent weight function for training data and an uncertainty score. The technique is evaluated favorably against prior works, both on synthetic data and on real datasets for many tasks including calibration, data with label noise and selective classification.

**Strengths:**

- The general premise of an all-purpose uncertainty evaluation tool, useful for both train and test time and adaptive to various sources of uncertainty, is a significant and original contribution that would be very useful
- The evaluation against prior works on a variety of uncertainty-related tasks is very comprehensive
- In section 4, the classifications into types 1-3 uncertainty and discussion of how instance weights should respond to these types was a useful step towards putting these different uncertainty problems under one theoretical framework

**Weaknesses:**

- Section 4's work with synthetic data has good potential to be interesting and illustrative of how ReVaR works differently in desired ways for different uncertainty types. However, I found it unclear how the "theoretical ideal for the instance dependent weights" listed in table 1 were determined. Some more information on their derivation might be helpful.
- Occasional minor notational things: Equation 3, definition of $\theta^*$, should $g_\Theta$ might instead be $g_{\Theta^*}$. The text following/explaining equation 6 introduces variables not used in equation 6; some minor rewriting could be useful here. Missing reference to a theory in section 4, scenario 1.

**Questions:**

- How were the targets in table 1 derived; why are these targets desirable?
- In distribution shift settings where we assume access to a validation set from the test distribution, is it not better sometimes to just fine-tune on the validation set (or a portion of it)? It could potentially be a good baseline comparison.

---

> ### Author Response · Authors · 2023-11-17
>
> Thank you for your thorough review, constructive feedback, and positive comments! We have addressed the raised questions and comments below. Since the reviewer assesses our work as having significant and original contributions with comprehensive evaluation, and has not expressed any major concerns with the work, may we request them to re-visit their rating for the work? We are of course happy to address any additional concerns or questions they may have.
>
> * *Section 4 … I found it unclear how the "theoretical ideal for the instance dependent weights" listed in table 1 were determined.*
>
> Since we control the amount and type of input-dependent noise we have introduced, we posit that the ideal weight should be proportional to the inverse of the introduced input-dependent noise [1] and directly proportional to the uncertainty due to covariate shift, i.e., $h:=(x-\mu)^2$ [2].  Apologies for the broken citation in the paper. In two-factor settings such as scenarios 2,4, we suspected that the weight would be a linear combination of the two contributing factors. Certainly, when either uncertainty from shift or label noise is increased while the other remains the same, we should expect the “ideal” weighing scheme to follow the larger noise source. So linear interpolation is a natural simple hypothesis.  Surprisingly, even this fairly rudimentary model is an excellent fit for the weights identified by our model; what's more, the different scenarios help dissociate the 3 methods -- loss-dependent reweighting (MWN), instance-based reweighting (our own novel baseline), and our full approach with variance-minimizing meta-regularization. For scenario 3, it is somewhat similar to scenario 1 as the missing features can be interpreted as labelled noise since their contribution cannot be modeled. Please refer to the updated draft for more details on deriving the targets for the scenarios 2 and 3. For scenario 5, since it is the shift in spurious features which do not play any role in deciding the output (given $W_e=0$), the weighting for all the instances should be uniform.
>
> * *Occasional minor notational things: Equation 3,6 [etc]. some minor rewriting could be useful here. Missing reference to a theory in section 4, scenario 1.*
>
> Thank you! These changes have been incorporated in the updated draft. Also, for eq. 3, we have added defined $\theta^*$, $\Theta^*$  and also it should be $g_\Theta$ only in eq. 3 as against $g_\theta^*$ since $\Theta$ is optimized in the outer loop in the exact bi-level formulation.
>
> * *In distribution shift settings where we assume access to a validation set from the test distribution, is it not better sometimes to just fine-tune on the validation set (or a portion of it)?*
>
> We clarify that any experiments we’ve shown for our method on domain shift data have used only ***unlabeled data*** from the test distribution, and not any labeled data whatsoever. Our approach can incorporate unlabeled data via the variance-minimization meta-loss (which is label-free). Fine-tuning, say, an ERM model in domain-adaptation settings is conventionally done using small amounts of ***labeled data*** from the test domain.
>
> Nevertheless, we did experiment with adding the same variance minimization loss as “fine-tuning” for an ERM model including unlabelled examples from the test domain in the validation set. The model is tuned with cross entropy loss for labelled examples and our unsupervised variance minimization objective for the unlabelled examples. We did this for two setups: first where train, val and test all belong to different domains (Chameleon, iWildcam) and second where train, val are from same domain but test from a different domain (DR country shift OOD). This method, labeled MCD(val-tuned) below, was not competitive. Please refer to appendix section J in the updated draft for more details.
>
> | | Camelyon | iWildCam | DR (OOD) |
> | --- | ----------- | ----------- | ------------ |
> | Revar | 76.32 | 77.98 | 89.94 |
> | Revar-PV | 78.12 | 79.86 | 91.23 |
> | MCD (val-tuned) | 75.25 | 76.34 | 88.06 |
>
> We would be happy to provide any further analysis.
>
>
> [1] Near Optimal heteroscedastic regression with symbiotic learning. COLT 2023
>
> [2] Direct importance estimation with model selection and its application to covariate shift adaptation. NeurIPS 2007.

---

> > ### Comment · Reviewer_1jLE · 2023-11-19
> >
> > Thank you for the detailed response; with these clarifications and revisions I will raise my review to a 6.

---

### Official Review · Reviewer_f1Z8 · 2023-10-31

**Soundness:** 2 fair
**Presentation:** 2 fair
**Contribution:** 2 fair
**Rating:** 6
**Confidence:** 3

**Summary:**

The paper proposes a method which learns a weighting function for the cross entropy loss, enabling re-weighting of the terms depending on how difficult (in an uncertainty sense) they are to classify. The method is learnt through a bi-level optimisation process, which is not dissimilar from a 'meta' training objective. The authors demonstrate their results on several datasets using ResNet-50 architectures.

**Strengths:**

The strengths of this paper are it's:
* Simplicity, the method seems very easy to implement and is intuitive to understand.
* The experiments are reasonably conclusive and operate on a significant number of datasets.

**Weaknesses:**

In terms of weaknesses:
* The paper seems quite rough, there are many undefined terms and functions ($R^2$, $g(x)$, $G$), missing citations (MMCE), lack of error bars in Table 4 - 7, etc.
* With a paper being set up the way it has, I would expect some proof, especially with the synthetic setup in Section 4
* Moreover, I'm not entirely convinced on why this method works, my understanding is that you are simply training the network to produce low variance in its predictions, which is manifested by the network essential learning dirac distributions over the parameters in dropout. It would be nice to see some investigation into this, or at least an explanation.
* Only performed on ResNet, there are many models available now which can be trained just as easily with similar compute.

**Questions:**

* What is the definition of $g(x)$? And how does this relate to $\theta$. I couldn't find this information when it was introduced, which made this paper very hard to grasp what was happening.
* How do $w_i$ and $\Theta$ relate?
* What is the theory in Scenario 1?
* What are the associated issues with this approach? I understand the objective of minimising uncertainty, but minimising the metric which provides the uncertainty is not the same thing. I'm concerned that all this is doing is simply collapsing the dropout distribution to all become the same parameters, i.e. it makes no difference on the prediction which parameters are selected.


Whilst I think there is some contribution in this paper, I just don't think in it's current form it's ready for publication. I would suggest that the authors improve:
* The quality of the paper to make it easier to understand what the method is, i.e. define $g(x)$ properly
* A proof would strengthen the paper significantly. If you can prove why minimising the variance provides improved uncertainty you're onto a winner.
* Add more architectures, using only ResNet-50 is not enough.
* I would also suggest removing the empirical evaluation on the top set up. If you have a toy set up like this, it should turn into a proof.

---

> ### Author Response · Authors · 2023-11-17
>
> * *The paper seems quite rough, there are many undefined terms and functions (R2,g(x),G ), missing citations (MMCE), lack of error bars in Table 4 - 7, etc.*
>
> We've addressed a number of the writing issues suggested by you and other reviewers. We thank the reviewer for the suggestions and questions;  we hope and believe the key contributions of our paper are now clearer, and that the writing does not detract from assessing the value / potential of our contributions. We have removed the notation G. R2 score is a popular metric used in statistics.
>  We have provided the updated tables 4-7 with std values in appendix section K. This is done due to limited space in the main paper. Also, please refer to Sec 5.1 regarding citations for MMCE and other baselines.
>
> * *Moreover, I'm not entirely convinced on why this method works, my understanding is that you are simply training the network to produce low variance in its predictions, which is manifested by the network essential learning dirac distributions over the parameters in dropout. It would be nice to see some investigation into this, or at least an explanation.*
>
> We apologize if the intuitions were not made clear. Here is a brief note, which we have incorporated into the paper as well; please let us know if this explanation addresses your concerns.
>
> The meta-regularizer has the following interesting properties by design: a) the objective is to minimize predictor model's variance on a *validation set unseen by the predictor model*, b) the mechanism for minimizing this variance is indirect, through reweighting the predictor model's training instances, c) the reweighting is accomplished by a meta-network.
>
> Thus, the meta-regularizer doesn't necessarily aim to produce a "more certain predictor" that may or may not be of good quality; instead, it forces the meta-network to upweight training instances that are most useful (informative) in reducing uncertainty. Conceptually, one can see that the ideal meta-network will need to weight training instances according to their input-dependent noise, or covariate shift, among other factors (see [1,2] for theoretical motivation of this claim). We ran experiments adding the variance minimization to the predictor's training loss; this produced neither a better predictor nor a better meta-network, showing that simply having a "certain" predictor isn't intrinsically valuable.  AUARC results on selective classification below (the new variant denoted with var_min_train):
> | 			    DR (in-Dist) |     DR (OOD) |  CIFAR-100 |       Im-100   |        Im-1k	|
> | -------------------------------- | -------------- | ------------- | -------------- | -------------- |
> Revar   	|	 94.12 土 0.1   |  89.94 土 0.1 |  93.20 土 0.1 |    94.95土0.1 |  88.20土0.2 |
> Revar (var_min_train) | 93.03 土 0.2 |     88.11 土 0.1 |  91.85 土 0.2 |    93.05土0.1 |  86.65土0.1 |
>
> We also ran experiments without this meta-learning framework, directly adding variance minimization alongside ERM in the paper (Sec. 5.1 under New baselines, Tables 2,3). This also leads to suboptimal performance.
>
> * *Only performed on ResNet, there are many models available now which can be trained just as easily with similar compute.*
>
> We've shown that fine-tuning a large-scale transformer model (PLEX) also shows similar gains on top of PLEX (Appendix Section C). This suggests that our ideas are valuable even for large-scale data and models. Please also refer to appendix E.1 where we experiment with ViT-Small architectures for both U-Scorer and Classifier and similarly RN-101. Also in appendix E.2 we evaluated WRN-28-10 architectures. The consistent gains suggest value across a range of architectures and sizes. If needed, we would be happy to provide more experimental results with various architectures in the paper.
>
>
> * *What is the theory in Scenario 1?*
>
> Apologies for the broken reference. Scenario 1 is inspired from a recent paper [1] which assumes a similar data model and establishes the optimality of importance weights that are inversely dependent to the noise. We have fixed this citation in the draft.
>
> * *What are the associated issues with this approach? I understand the objective of minimising uncertainty, but minimising the metric which provides the uncertainty is not the same thing.*
>
> As described above, the uncertainty-minimizing regularizer is a tool to help the meta-network to focus on relevant training instances, and thereby capture notions of instance-based uncertainty. Table 1 clearly shows that the regularization is what allows our method to accurately capture input-dependent noise, across a range of scenarios.
>
>
> [1] Near Optimal heteroscedastic regression with symbiotic learning. COLT 2023

---

> ### Author Response · Authors · 2023-11-17
>
> * *What is the definition of g(x)? And how does this relate to \theta. I couldn't find this information when it was introduced, which made this paper very hard to grasp what was happening.*
> * How do  $w_i$ and $\Theta$ relate?*
>
> We sketch the proposal briefly below; apologies for the lack of clarity in our original draft.
>
> 1. We introduced the basic goal in Sec3.1 – we want to learn two functions  $f_\theta(x)$ (classifier) and $g_\Theta(x)$ (an uncertainty measure associated with f that maps an input to a number [0 1]). Their parameters are ($\theta$, $\Theta$) respectively.  We can define g(x) functionally, i.e., in terms of how we want the output to behave. One example is that it maximizes accuracy of a given f(x) for a given budget (eq.1) , i.e., that it correlates strongly with the likelihood that the classifier is correct.
>
> 2. **We leverage previous work in learned reweighting of training data (eq.2) which learns free parameters $w_i$ for each training instance $x_i$. We believe that these weights $w_i$ are fairly good stand-ins for instance hardness, since upweighting hard samples is known to be beneficial.**  But we want to be able to score new, unseen test instances, so instead of w_i we learn a neural network $g_\Theta(x_i)$ (Section 3.3).  In other words, we replace $w_i$ in Eq. 2 with $g(x_i)$ (Eq. 3).
>
> 3. Finally, although we have now enabled test-time applications with point #2, the standard bilevel objective of Eq.3 is insufficient, so we add a variance-minimization regularizer (section 3.3). Please see “Intuition” in the updated draft for why we chose to do this, and  Section 4 / Table 1/ Ours vs IBR for empirical evidence across scenarios that our contribution #3 improves significantly over our contribution #2 above.
>
> We believe perhaps point #2 above, **especially the text in bold**, was insufficiently explained, perhaps leading to confusion. Hopefully this helps clarify, and we have also made changes in the text to reflect this.
>
>
> -  *I would suggest that the authors improve:*
>      - *The quality of the paper [...]*
>
>      - *A proof would strengthen the paper significantly. [...]*
>
>      - *Add more architectures, using only ResNet-50 is not enough.*
>
>      - *I would also suggest removing the empirical evaluation on the top set up. If you have a toy set up like this, it should turn into a proof.*
>
> 1. We have provided a conceptual sketch of the design, and why we expect minimization of variance to work. We have rewritten various sections incorporating other feedback from you and other reviewers w.r.t. the presentation.  As noted above, we have already shown results on large-scale pretrained transformer models (PLEX, Appendix section C) and analyzed other architectures including ViT-S, RN-101 and WRN-28-10.
>
> 2. Removal of empirical evaluation (Section 4): We respectfully disagree with this suggestion for the following reasons: a) the synthetic setup allows us to exactly control the type and degree of noise introduced, b) using these controls, we are able to clearly dissociate (table 1) what can be achieved using loss-based, input-based (our own baseline), and meta-variance-reduction-based (our full method) reweighting. It establishes not only ordering between these methods with clear large gaps (table 1), but also the ability of our method to handle gradations of contribution input noise source  (figure 1). c) we have also attempted to connect the real-world experiments to each of these scenarios, showing that our gains in synthetic settings generalize.
>
> 3. Proofs: As mentioned above, our approach smoothly interpolates between two sources of noise, when their contributions are varied. [1,2] provide theoretical justifications for the target weightings in Table 1, and do so by explicitly estimating the uncertainty measure (importance sampling ratio in the latter and label noise variance in the former) for calculating weights. Our meta-network does this implicitly without any estimation, and can moreover also interpolate. Theoretical justification of these empirical findings will therefore need to integrate a number of scenarios & corner cases requiring very different ideas spanning almost a decade of work, alongwith the challenges of analyzing the bi-level training dynamics; it is hence out of scope for this paper although an active area of work for us.
>
> [1] Near Optimal heteroscedastic regression with symbiotic learning. COLT 2023
>
> [2] Direct importance estimation with model selection and its application to covariate shift adaptation. NeurIPS 2007.

---

> ### Author Response · Authors · 2023-11-21
>
> * **Additional data showing replication of all results in table 1 for ViT-Small/WRN-28-10 architecture.**
>
> As discussed above, we have already conducted experiments with various architectures for classifier and U-Score in appendix F.1 and F.2 (E.1, E.2 in the original draft). However, given the reviewer's concerns, we provide further results replicating table 1 for the ViT-Small and WRN-28-10 architectures for the classifier in appendix section L in the updated draft. The performance gains either have a minor increase or are same as Table 2 in the main paper when using these architectures. This further verifies the effectiveness of our method and advocates for its usefulness for various tasks requiring different kinds of neural network architectures.
>
> We're happy to address any additional questions or concerns; we request the reviewer to consider our responses above.

---

> > ### Comment · Reviewer_f1Z8 · 2023-11-22
> > **Response**
> >
> > Thank you for a detailed response and addressing the concerns I had, it's highly appreciated. I feel many of them stemmed from the original lack of clarity in the draft. Having seen the new draft had my initial concerns addressed I have decided to raise my score accordingly.

---

### Official Review · Reviewer_bRmp · 2023-11-10

**Soundness:** 4 excellent
**Presentation:** 2 fair
**Contribution:** 3 good
**Rating:** 8
**Confidence:** 3

**Summary:**

In this paper, the authors propose an instance weighting methodology based on an auxiliary network that quantifies the uncertainty of the predictor subject to learning. More precisely, a bilevel optimization formulation enables one to simultaneously learn the predictor and the auxiliary uncertainty quantifier, whereas a meta-level loss enforces to reduce the predictor’s uncertainty by minimizing a variational approximation of Bayesian Neural Networks by means of multiple inference passes employing Dropout regularizations. As the authors show, the method proves to be an effective means to achieve a more uncertainty-aware, and consequently better generalizing predictors.

**Strengths:**

- Strong empirical evidence of the reasonability of the method. Also, the experiments are at an impressive scale, supporting the claims of the work.
- Rich set of recent baselines considered in the experiments, fair comparisons.
- I really like that this method allows for augmenting previous robust approaches, e.g., PLEX, with the proposed solution. That significantly boosts its applicability.
- Computational efficiency concerns are thoroughly addressed in the appendix.

**Weaknesses:**

Major:

- The contribution and distinctions to previous works (which e.g. also use a bilevel optimization formulation for reweighting) should be made more clear in the thread of Section 3. Often, it does not become clear when something is new in the course of the ReVaR proposal, and when previous works are revisited.
- Many grammatical and orthographic issues, e.g., missing articles (“*the* dataset” → motivational questions in Section 1), misplaced words (“captures captures” → beginning of Section 5), …
- Section 4 is very hard to read, some design choices also appear arbitrary. For instance, intuitively, what is the role of $X_c$, $X_e$, $W_c$ and $W_e$, and why are the dimensionalities chosen as described in the paper? Elaborating more on the setup would help to better understand the settings and allow for estimating the significance of these results.

Minor:

- Sometimes unclear / inconsistent notation, e.g., “p=w=g(x)” in  Section 1.1, also see my previous comment on $X_c$, $X_e$, $W_c$ and $W_e$.
- As far as I can see, there is no code served along with the paper, which does not support the reproducibility of this work.
- Section 4, Scenario 1: „?“ Broken reference
- Details about the U-score model architecture appear in the appendix only – as this is not as trivial as “normal” predictors, this should be already hinted at in the main paper.

**Questions:**

1. While Section 4 gives a comprehensive overview of different forms of captured uncertainties in $g_\theta$, I would be also interested in seeing concrete learning behaviors of $g$ in the real-world experiments. Perhaps the authors could augment the results by showing patterns of $g$ in this regime, e.g., by plotting the distributions of the learned weights, and how these evolve over the training. Right now, it is hard to get an impression about the learning dynamics, raising the following questions: Does the novel meta loss slow down the training by making it more cautious, or does it even accelerate the training? Note that I am not referring here to what has been discussed in E.3, but with the focus on learning curves. For instance, setting the maximum number of epochs / data points to control training costs is often a critical consideration in real-world application at larger scales.
2. Could the U-SCORE and predictor weights be shared? How would this affect the training?

---

> ### Author Response · Authors · 2023-11-17
>
> Thank you for your thorough review, constructive feedback, and positive comments! A few brief answers follow, and we have incorporated the suggestions into an updated draft.
>
> * *contribution and distinctions to previous works …in the thread of Section 3*
>
> Apologies, we will make this more clear. The 2 key innovations are instance-conditioning (previous work learned either free parameters or functions of instance loss) and the novel meta-regularizer of variance minimization. Instance conditioning allows us to score unseen test instances (without access to labels), and thereby enables new use-cases such as selective classification that the previous methods cannot do. Variance minimization allows us to capture a variety of input-dependent uncertainty measures  (Table 1) that loss-conditioning (MWN) and instance conditioning (IBR, our own contribution) alone cannot.
>
> * *Many grammatical and orthographic issues*
>
> Thank you; we are incorporating the suggested changes along with a round of proofreading.
>
> * *Section 4 is very hard to read, some design choices also appear arbitrary. what is the role of Xc,Xe,Wc,We,  and , and why are the dimensionalities chosen as described in the paper?*
>
> We apologize for the lack of clarity. The variables are introduced as a general setup for introducing specific sources of input-dependent noise as described in the scenarios 1-5. These scenarios are often studied in isolation; we wanted to evaluate methods across these scenarios as a way of dissociating the contributions of each method for each scenario.  We have updated the writing to make the above and other aspects clear.
>
> (Briefly, the [Xc,Xe] partitioning exists to enable scenarios 3, 4 where some of the features of the underlying generative model are not available to the learner.  In these scenarios, although the label is still generated as a function of the entire input, the learners are only provided Xc as input. The contributions of Xe to the label, being inaccessible to the learner, essentially function as input-dependent noise through the conditional dependence Xe|Xc.)
>
> The design choices  (dimensionality etc) do not materially affect the findings; we verified this by running experiments on additional configurations for synthetic data with randomly chosen dimensions (in the range 50-200), and all of them showed similar findings. We can share the results if needed.
>
>
> * *Sometimes unclear / inconsistent notation. Section 4, Scenario 1: „?“ Broken reference. Details about the U-score model architecture appear in the appendix only*
>
> Thank you; we have incorporated the suggested changes along with a round of proofreading. Also we have added a line at the end of section 3 that our meta-network is also implemented as standard neural network architecture like the classifier.
>
>
> * *As far as I can see, there is no code served along with the paper, which does not support the reproducibility of this work.*
>
> We will be releasing code along with the final version of the paper if accepted.
>
>
> * *it is hard to get an impression about the learning dynamics [...]*
>
> This is a very interesting point; we examined the training dynamics but did not see any overt evidence of, e.g., faster convergence for our approach compared to ERM. We are performing additional analyses to be shared soon.
>
> * *Could the U-SCORE and predictor weights be shared? How would this affect the training?*
>
> We considered an architecture where the meta-network was k fully-connected layers  (for some values of k) on top of the  primary classifier’s final layer representation. The meta-objective only updated the k layers, and not the shared encoder (otherwise, training was not stable).  Looking at AUARC in selective classification, the shared architecture performs worse than a separate meta-network.
>
>
>
> |   | DR (ID) | DR (OOD) | CIFAR-100 | ImageNet-100 | ImageNet-1K |
> | ------ | ------ | ------ | ------ | ------ | ------ |
> | Ours | 94.12 $\pm$ 0.1 | & 89.94 $\pm$ 0.1 | & 93.20 $\pm$ 0.1 |  & 94.95 $\pm$ 0.1 | & 88.20 $\pm$  0.2 |
> Ours (Shared, K=4) | 93.05 $\pm$ 0.1 | 88.07 $\pm$ 0.1 | 92.03 $\pm$ 0.1 | 93.67 $\pm$ 0.1 | 86.85 $\pm$  0.1 |
> Ours (Shared, K=6) | 93.12 $\pm$ 0.1 | 88.13 $\pm$ 0.1 | 92.16 $\pm$ 0.1 | 93.32 $\pm$ 0.1 | 86.75 $\pm$  0.1 |
> Ours (Shared, K=8) | 92.98 $\pm$ 0.1 | 88.02 $\pm$ 0.1 | 91.98 $\pm$ 0.1 | 93.10 $\pm$ 0.1 | 86.68 $\pm$  0.1 |
>
> A possible explanation is that the classifier needs to find discriminative features, whereas the  meta-network needs to identify features, potentially across classes, that flag instance hardness; hence they may need different representations.  We also tried other configurations where an earlier-layer representation from the primary model was input to the meta-network, but no improvements were observed. We have added this in the updated draft (appendix section I)

---

> > ### Comment · Reviewer_bRmp · 2023-11-22
> >
> > I want to thank the authors for this thorough rebuttal, which adequately addresses my concerns. After checking the other reviewers' comments, I do not see any critical weaknesses I have missed and which could be a profound reason for rejection, so I keep my current scoring, leaning towards acceptance.

---

### Author Response · Authors · 2023-11-17

We thank the reviewers for their thoughtful comments. Below we have addressed some of concerns regarding our work. Also, we have uploaded a new draft, and the primary changes are responses to reviewer comments for clarity in writing, and additional experiments (appendix).

---

### Meta-Review · Area_Chair_Wcbs · 2023-12-09

**Metareview:**

The authors propose a bilevel optimization problems, in which a predictor is learned simultaneously with an auxiliary uncertainty quantifier, which is used for instance weighting. A meta-level loss enforces reduction of the predictor’s uncertainty by minimizing a variational approximation of Bayesian Neural Networks by means of multiple inference passes employing Dropout regularizations. The authors show their method to be effective in achieving uncertainty-awareness and better generalisation performance.

Although the reviewers raise a number of critical points in their original reports, there is agreement that the paper holds promise, and the authors' idea of the bilevel approach looks quite intriguing. The authors also showed a high level of commitment during the rebuttal phase and did their best to respond to the comments and to improve the submission. This was appreciated and positively acknowledged by all. In the discussion between authors and reviewers, essentially all critical points could be resolved.

**Justification For Why Not Higher Score:**

Good paper, but not outstanding.

**Justification For Why Not Lower Score:**

Good enough to justify acceptance.

---

### Decision · Program_Chairs · 2024-01-16

Accept (poster)